# SHAPE-ADAPTIVE GUIDANCE SIGNAL FOR INTERACTIVE CORTICAL SULCAL LABELING

## ABSTRACT

Image segmentation is a fundamental task in image data analysis that assigns semantic labels to enhance the understanding of imaging data. In the context of neuroimaging data, the accurate labeling of cortical sulci is crucial for providing deep insights into the link between cortical folding patterns and cognitive functions. Yet, fully automatic methods often struggle with labeling small and shallow sulci due to their high anatomical variability and the scarcity of annotated training data. In this context, interactive segmentation may offer a promising alternative by incorporating minimal human input to refine labels. However, recent learning-based interactive approaches often rely on 2D projections of surface data, typically designed for generic and relatively small 3D meshes. This dimensional simplification inherently limits their ability to capture subtle folds and deeply buried structures of cortical surfaces. In this paper, we introduce a shape-adaptive guidance signal for interactive cortical sulcal labeling using spherical convolutional neural networks. Thanks to the use of spherical mapping, our approach preserves structural information without the need for sacrificing anatomical details. To effectively encode user clicks along cortical folding patterns, we solve the eikonal equation with a speed function that incorporates the mean curvature of the cortical surface unlike conventional encoding schemes using equidistance. This curvature-aware signal captures fine-grained anatomical details to guide the neural network focus on the intended refinement. Experimental results on 72 subjects with 17 sulci on the lateral prefrontal cortex show that even a single click using the proposed encoding scheme outperforms fully automatic methods and equidistance schemes, while achieving both efficiency and improved labeling accuracy.

## 1 INTRODUCTION

Image segmentation is one of the fundamental tasks in computer vision to assign a semantic label to each pixel or regions of interest (ROIs). This forms the basis of a wide range of applications such as object recognition or scene understanding in natural images, as well as disease diagnosis in medical image analysis (Long et al., 2015; Ronneberger et al., 2015; Cordts et al., 2016). While fully automatic methods have been largely explored by the previous studies, the resulting labels often require user validation and manual correction especially in precision-critical scenarios such as clinical applications (Maleike et al., 2009; Fournel et al., 2021; Spaanderman et al., 2025).

Recent segmentation studies have shown that geometric approaches naturally capture intrinsic geometric properties and generalize across irregular domains unlike traditional deep learning methods that rely on grid-like data formats. These strengths make them particularly effective for tasks involving structural understanding and spatial awareness. While many geometric deep learning methods employ graph convolutional neural networks (CNNs) (Kipf & Welling, 2017; Hamilton et al., 2017; Velickovic et al., 2017) for broad applicability, specific domains often exhibit unique geometric characteristics that necessitate specialized adaptations. For instance, spherical topology (continuous, closed surface and rotational symmetry) poses unique challenges that may not be fully addressed by traditional graph CNNs designed for discrete and irregular graph structures. Spherical CNNs perform convolutions directly on the sphere, enabling rotation-equivariant and geometrically consistent feature extraction. By respecting the continuous nature of the spherical surface and its inherent rotational symmetry, spherical CNNs enable more accurate and consistent learning, particularly in applications where orientation and global structure are critical such as neuroimaging (Seong et al.,

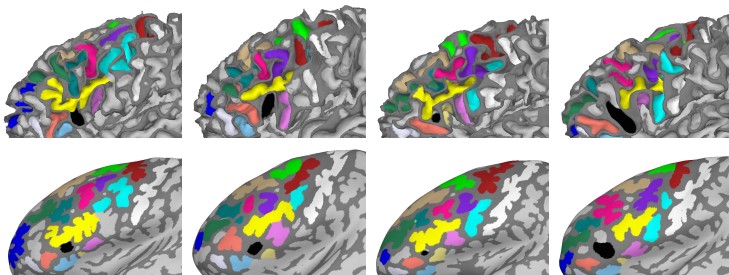

Figure 1: Anatomical variability in lateral prefrontal cortex. Manually identified sulci are overlaid on the white (top) and inflated surfaces (bottom). Compared to large and deep sulci, small and shallow sulci exhibit high anatomical variability across individuals. This variability makes them particularly challenging to label accurately and difficult to localize using fully automatic methods.

2018; Zhao et al., 2019; Parvathaneni et al., 2019; Lyu et al., 2021; Ha & Lyu, 2022), climate modeling (Weyn et al., 2020; Defferrard et al., 2021), and panoramic vision (Jiang et al., 2019; Yu & Ji, 2019). Particularly, spherical CNNs are becoming increasingly valuable in cognitive neuroscience as a powerful tool for advancing the foundational goal of understanding the relationship between neuroanatomy and cognitive functions.

Cortical sulci, the indentations of the cerebral cortex in gyrencephalic brains, play a crucial role in shaping its structural and functional organization. Traditionally, studies have focused on large and easily identifiable sulci to investigate cortical folding patterns. However, recent research has shifted toward examining finer-scale morphology of smaller and shallower sulci, while revealing their potential relevance to higher-order cognitive functions such as reasoning, working memory, and abstract thought (Voorhies et al., 2021; Willbrand et al., 2022; Yao et al., 2023; Willbrand et al., 2023; 2024). Such growing focus highlights the critical role of sulcal anatomy in understanding the neural basis of cognition. Thus, improving labeling techniques is essential for accurately capturing these finer details and advancing our understanding of cortical complexity.

To date, spherical CNNs have demonstrated strong performance in processing cortical data as an invertible spherical mapping is readily available (Fischl, 2012). Yet, accurately labeling small and shallow sulci in a fully automatic manner remains a significant challenge. As illustrated in Figure 1, these sulci exhibit large anatomical variability in location and shape across individuals. Small and shallow sulci are occasionally - but not consistently - adjacent to each other within a single connected component. Moreover, the limited availability of manually labeled data complicates model training. Although several spherical CNNs-based techniques have shown promise for fully automatic sulcal labeling (Hao et al., 2020; Lyu et al., 2019; Lee et al., 2025a;b), shallow sulci continue to require expert validation and correction due to their subtle and inconsistent morphology. Consequently, cortical sulci are still manually corrected by trained raters in neuroscience studies. This labor-intensive process restricts the scalability of studies and thus highlights the urgent need for reliable identification tools under minimal user supervision.

Interactive segmentation bridges the gap between manual labeling and automatic methods. In this paradigm, sparse user inputs such as clicks, scribbles or bounding boxes are used to guide the segmentation process. These interactions can allow for precise and efficient delineation under minimal supervision. The approach is particularly valuable in domains where manual annotation is costly and boundary definitions are ambiguous, hence requiring expert validation. While interactive frameworks have been successfully developed for 2D images and volumetric data in Euclidean spaces (Jang & Kim, 2019; Lin et al., 2020; Diaz-Pinto et al., 2022), their extension to non-Euclidean representations remains relatively underexplored. Given that many real-world data modalities such as 3D point clouds and surface meshes are inherently unstructured, this motivates a shift toward geometric deep learning to complex data representations beyond traditional grid-based formats.

Traditional mesh-based interactive segmentation methods like graph cuts (Boykov & Jolly, 2001; Fan et al., 2011) and harmonic fields (Au et al., 2011; Zheng et al., 2011) offer efficient optimization and smooth boundary generation. While effective for coarse segmentation tasks, they often fail to capture fine-grained geometry like cortical folds due to over-smoothing and limited sensitivity to

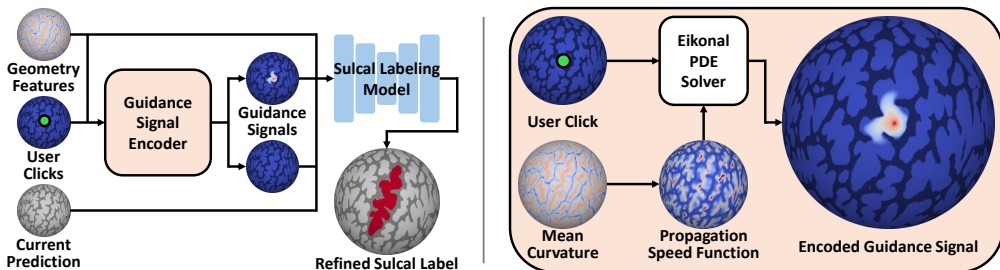

Figure 2: An overview of the proposed interactive labeling model (left) and guidance signal encoder (right). The model uses the mapped geometry features, positive or negative clicks encoded as two-channel guidance signals, and an optional current prediction. The guidance signals are formulated via the eikonal equation that incorporates local folding patterns. The model performs initial labeling and refines it iteratively with subsequent user clicks.

anatomical details. Recent learning-based approaches on 3D point clouds (Kontogianni et al., 2023) and meshes (Lang et al., 2024) increasingly utilize pretrained 2D foundational models such as the Segment Anything Model (SAM) (Kirillov et al., 2023). The core idea underlying this strategy is to project the 3D surface mesh onto multiple 2D planes to provide segmentation cues from different viewpoints. Despite their potential, applying these models to cortical surface data remains non-trivial as it necessitates projecting the 3D mesh onto a 2D plane, which often results in the occlusion of deeply buried brain structures such as the Silvian fissure.

Spherical mapping may offer a promising solution to the challenges posed by buried brain structures, as it enables user inputs to be encoded directly as spherical signals without the need for planar projection and is naturally compatible with spherical CNNs. To fully exploit the framework for interactive segmentation, however, it is essential to design guidance signals that effectively conveys user intent on the sphere since their representation has a pivotal impact on segmentation performance (Marinov et al., 2023). A central challenge lies in effectively encoding these signals on the sphere. To this end, a straightforward approach involves applying a Euclidean distance transform that models user interaction based solely on spherical spatial proximity (i.e., geodesic disk) centered at the click point. However, this approach tends to overlook underlying anatomical structures, which may result in suboptimal performance particularly in the context of fine-grained sulcal labeling. To the best of our knowledge, no prior studies have investigated interactive geometric segmentation methods that explicitly incorporate surface geometry to generate structure-aware guidance signals.

In this paper, we propose a novel shape-adaptive guidance signal for interactive cortical sulcal labeling. To overcome the limitations of buried anatomical representation, we employ a spherical mapping of cortical surfaces to enable user interaction to be performed directly on the spherical domain. The user interaction (i.e., click point) is then generated by solving the eikonal equation with isotropic speed, which yields faster propagation along sulcal valleys to effectively capture fine-grained anatomical characteristics. The proposed method further supports iterative refinement by modeling subsequent user clicks from expert annotators. In our experiments, we validated the effectiveness of the proposed method on a dataset comprising 72 healthy subjects annotated with 17 sulcal labels on the lateral prefrontal cortex (LPFC). The proposed method achieves higher accuracy by just a single click against automatic labeling methods. Compared to other simple encoding strategies, the proposed encoding scheme yields higher initial accuracy and enables subsequent performance gains following the first user click. Figure 2 illustrates a schematic overview of the proposed framework.

## 2 METHODS

### 2.1 PROBLEM STATEMENT AND METHODOLOGICAL OVERVIEW

Our interactive sulcal labeling model is trained to perform binary segmentation of a cortical surface for individual sulci with support for iterative refinement. Although cortical surfaces contain multiple sulci, incorporating multiple sulci in interactive segmentation can introduce several challenges. These include an increased number of possible user click sequences, arbitrary shifts in user focus

across sulci, and difficulty in ensuring balanced training for each sulcus. While a few previous multi-object interactive segmentation studies (Rana et al., 2023; Yue et al., 2024) have discussed these challenges, a widely-adopted protocol for multi-object segmentation has not yet been established. In addition, we train a separate supervised model for each sulcus to account for the distinct morphological characteristics of LPFC sulci, including size, branching pattern, and overall shape. This per-sulcus modeling approach is consistent with common practices in medical image interactive segmentation (Wang et al., 2018; Luo et al., 2021; Diaz-Pinto et al., 2022) and allows each model to specialize in the characteristics of each individual sulcus.

Since the cortical surface is genus-zero, its geometric features can be mapped onto the unit sphere. This simplifies our problem on a spherical domain. For each user interaction at a spherical location, the labeling model takes a $K$-dimensional geometric feature vector, a current prediction, and encoded user guidance signals, as shown in Figure 2. The model output is thus a binary discriminant function $\mathcal{F} : \mathbb{R}^K \to \mathbb{R}^2$ to infer labels $z \in \{0, 1\}$ at each spherical location $\mathbf{x} \in \mathbb{S}^2$.

We optimize $\mathcal{F}$ using a spherical CNN to support iterative refinement based on subsequent user inputs with interaction generated by a simulated click sampler in a supervised manner. At each iterative step, we sample a click from mislabeled regions between the manual label and the current prediction to mimic a trained rater refining both over- and under-segmentation. Hence, $\mathcal{F}$ is continuously refined with the labeling results from previous steps. In the following sections, we will describe each component of the proposed framework.

## 2.2 ITERATIVE USER CLICK SIMULATION

During training and evaluation, we simulate user interactions by generating both positive clicks on under-segmented regions and negative clicks on over-segmented regions to correct the current prediction. Unlike existing studies that simulate multiple clicks at once (Xu et al., 2016; Wang et al., 2018; Luo et al., 2021), we adapt an iterative click simulation strategy (Mahadevan et al., 2018; Sofiiuk et al., 2022) that simulates one click at a time and updates the label each time. This design allows the model to provide immediate feedback after each click and to progressively refine remaining errors with each subsequent click. We further introduce spatial variability to the click simulation to account for slight differences in click locations. Each simulated click is encoded as a separate input channel as illustrated in Figure 2, which enables the model to explicitly distinguish positive and negative interactions.

Specifically, we simulate user interactions through the following steps. First, the manual label is compared with the current prediction to identify the largest mislabeled connected component from either a false positive or false negative region. If no current prediction is available (i.e., during the initial click), the largest connected component is extracted directly from the manual label. Once the largest mislabeled component is identified, geodesic distances from its boundary to interior points are computed on the sphere. Here, points with distances below the median within the component are filtered out to avoid sampling too close to the boundary. This operation can adaptively handle mislabeled regions of varying sizes: small regions have only a few points filtered out, while large regions have more. Weighted random sampling is then performed based on the computed distance. Specifically, the distances of the interior points are normalized by their maximum distance, and softmax is applied to assign a weight to each point. In this way, iterative clicks can be simulated near the region center with modest variation. This mimics the human annotator refining the largest mislabeled region by clicking near its center.

## 2.3 GUIDANCE SIGNAL ENCODING

In CNN-based interactive segmentation, user interactions are encoded as explicit guidance signals on the spatial domain. The guidance signal directly cues the model to refine inaccurate regions of the current prediction. It is important to note that the guidance signal is not intended to mark correct regions but rather provides directional cues that guide the model's attention. The model refines the current prediction by jointly incorporating the guidance signal, the current prediction itself, and the imaging data. Given the use of a spherical CNN in our study, guidance signals need to be mapped onto a spherical domain to align with the model's native coordinate system. To this end, we investigate two approaches: equidistance-based and shape-adaptive signals. Figure 3 illustrates example guidance signals.

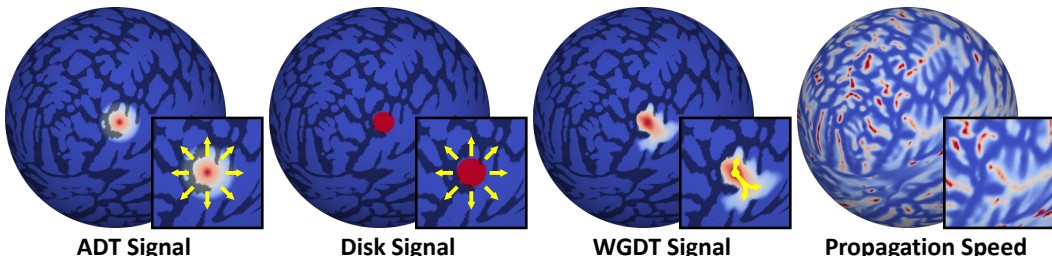

**ADT Signal**   **Disk Signal**   **WGDT Signal**   **Propagation Speed**

Figure 3: Guidance signals. The ADT and Disk signals are based on equidistance, while the WGDT signal adapts to cortical folding patterns, propagating faster along sulci and slower in gyri. The WGDT signal tends to remain localized along folds unlike the ADT and Disk signals. Cortical folding patterns are overlaid to each of the signals for better visual inspection.

### 2.3.1 ANGULAR DISTANCE TRANSFORM

Encoding clicks using a distance transform is a popular approach in CNN-based interactive segmentation studies (Xu et al., 2016; Li et al., 2018; Hu et al., 2019). To extend the Euclidean distance transform from 2D images and 3D volumes to the spherical domain, we encode a click $\mathbf{c} \in \mathbb{S}^2$ using an angular distance transform (ADT). At $\mathbf{x} \in \mathbb{S}^2$, we ensure that the signal is bounded in the range $[0, 1]$ and assigns higher weights to locations closer to $\mathbf{x}$.

$$\text{ADT}(\mathbf{x}, \mathbf{c}, \sigma) = \begin{cases} 1 - \frac{1}{\sigma} \arccos\left(\mathbf{x} \cdot \mathbf{c}\right), & \text{if } \arccos\left(\mathbf{x} \cdot \mathbf{c}\right) \leq \sigma \\ 0, & \text{otherwise.} \end{cases} \tag{1}$$

### 2.3.2 BINARY DISK

Another commonly used signal is the small-radius binary disk (Benenson et al., 2019; Sofiiuk et al., 2022). This signal is defined as a binary mask that covers all points within an arc-length (or angle) $\sigma$ from a user click $\mathbf{c}$. This mask can be defined straightforwardly as

$$\text{Disk}(\mathbf{x}, \mathbf{c}, \sigma) = \begin{cases} 1, & \text{if } \arccos(\mathbf{x} \cdot \mathbf{c}) \leq \sigma, \\ 0, & \text{otherwise.} \end{cases} \tag{2}$$

### 2.3.3 WEIGHTED GEODESIC DISTANCE TRANSFORM

We hypothesize that adequately covering the target sulcal region in the guidance signals helps the model in labeling and refining the sulci. However, the ADT-based guidance signals rely solely on the angular difference between a user click and points on the unit sphere. This approach overlooks the intrinsic morphology of the original cortical surface (see Figure 3). Consequently, these signals may inadequately cover the targeted region or excessively affect neighboring areas, which yields suboptimal refinement. A few studies have proposed geodesic distance transform over underlying imaging data on 2D or 3D grids (Wang et al., 2018; Luo et al., 2021). These approaches define the path cost as a function of pixel or voxel intensity differences. Yet, for geodesic transforms on a 3D mesh to be effective and to properly align the guidance signal, the surface geometry must be explicitly considered.

To address the issue, we propose a novel guidance signal of weighted geodesic distance transform (WGDT) by performing wavefront propagation on the unit sphere from a click $\mathbf{c}$, with the propagation speed function designed for sulcal labeling. Let $\mathbf{c}$ be a seed point. Following Sethian & Vladimirsky (2003), the minimum travel-time (equivalently, distance) $u_{\mathbf{c}}(\mathbf{x})$ from $\mathbf{c}$ to $\mathbf{x} \in \mathbb{S}^2$ meets the propagation equation for $\exists F \in \mathbb{R}^+$:

$$\|\nabla u_{\mathbf{c}}(\mathbf{x})\| F\left(\mathbf{x}, \frac{\nabla u_{\mathbf{c}}(\mathbf{x})}{\|\nabla u_{\mathbf{c}}(\mathbf{x})\|}\right) = 1, \quad \mathbf{x} \in \mathbb{S}^2$$
$$u_{\mathbf{c}}(\mathbf{x}) = 0, \qquad\qquad\qquad \mathbf{x} \in \mathbf{c} \tag{3}$$

where $F$ is a positive real-valued propagation speed function. In our problem setting, $F$ is considered an isotropic function known as the eikonal equation. This equation describes wavefront propagation

with a constant speed in all directions. We design $F$ as a spherical function representing the mean curvature derived from the cortical surface $H : \mathbb{S}^2 \to \mathbb{R}$, so that the signal propagates faster along sulcal regions ($H \geq 0$) and slower along gyral regions ($H < 0$).

$$F\left(\mathbf{x}, \frac{\nabla u_{\mathbf{c}}(\mathbf{x})}{\|\nabla u_{\mathbf{c}}(\mathbf{x})\|}\right) = e^{kH(\mathbf{x})}, \tag{4}$$

where $k \in \mathbb{R}^+$ is a hyperparameter that modulates the influence of $H$ on the propagation speed. Once $F$ is computed, we clamp $F$ within the range $[0.05, 10]$ to mitigate propagation instability. We utilize the fast marching algorithm (Sethian, 1996) for solving the eikonal equation on a discrete surface. After solving Equation 3, the WGDT signal is given by

$$\text{WGDT}(\mathbf{x}, \mathbf{c}, \sigma) = \begin{cases} 1 - \frac{1}{\sigma} u_{\mathbf{c}}(\mathbf{x}), & \text{if } u_{\mathbf{c}}(\mathbf{x}) \leq \sigma \\ 0, & \text{otherwise,} \end{cases} \tag{5}$$

where $\sigma$ is the maximum travel time of the propagation to localize the feedback. Similar to the ADT signals, higher weights are assigned to locations closer to $\mathbf{x}$.

## 2.4 ITERATIVELY WEIGHTED SULCAL LABELING LOSS

Following the previous studies on cortical sulcal labeling (Lyu et al., 2021; Lee et al., 2025a), we use cross-entropy loss to minimize the error between ROI-wise probability $p$ given by $\mathcal{F}$ at the $i$-th iteration step and the manual sulcal definition $z$:

$$\mathcal{L}_{\text{label}}^i = -\sum_{n \in \{0,1\}} \log(p_n, z_n). \tag{6}$$

To guide the model to learn iterative refinement over clicks, we adapt iterative click loss (ICL) proposed by (Sun et al., 2024). This approach embeds the number of clicks in the loss by imposing larger weights to the losses from later steps.

$$\mathcal{L}_{\text{ICL}} = \sum_i \beta_i \mathcal{L}_{\text{label}}^i, \tag{7}$$

where $\beta_i \in \mathbb{R}^+$ is the weight factor for the $i$-th step in accumulating the losses.

## 2.5 BACKBONE ARCHITECTURE

We employ SPHARM-Net (Ha & Lyu, 2022) as our backbone model. By utilizing spherical harmonics-based convolutions, SPHARM-Net inherently achieves rotational equivariance, where output features transform predictably under input rotations. This property reduces the need for extensive data augmentation without loss of performance. However, achieving rotational equivariance comes at the cost of expressive power in SPHARM-Net due to the isotropic weighting of its convolutional filters. This may limit the model's ability to capture fine-grained cortical structures. It is noteworthy that the proposed guidance signal addresses this limitation by complementing the extracted features, which provides additional cues to refine sulcal labeling.

## 3 EXPERIMENTAL SETUP

### 3.1 IMAGING DATA

We used a subset of the Human Connectome Project (HCP) (Van Essen et al., 2012). 72 participants were randomly chosen (aged 22-36 years old; 36 males and 36 females). T1-weighted scans were acquired in native space from the HCP dataset (Glasser et al., 2013). We used geometric features (Fischl, 2012): mean curvature of the white-matter surface (*curv*), average convexity (*sulc*), and mean curvature of the inflated surface (*inflated.H*).

Following the most recent definitions of LPFC sulci (Petrides, 2018; Voorhies et al., 2021; Willbrand et al., 2022; Yao et al., 2023; Willbrand et al., 2023; 2024), we manually defined 8 large, consistent sulci and 9 smaller and more variable sulci on the left hemisphere of each subject, as in Table 1. Detailed definitions and positional relationships of each sulcus are provided in Appendix A.2.

Table 1: Color-coded acronyms of large and small sulci on lateral prefrontal cortex.

| large, consistent | | | | | small, variable | | | | |
|---|---|---|---|---|---|---|---|---|---|
| cs | ⬜ | sprs | 🟥 | iprs | 🟦 | pmfs-p | 🟪 | pmfs-i | 🟥 | pmfs-a | 🟩 |
| ifs | 🟨 | sfs-p | 🟫 | sfs-a | 🟩 | ds | 🟪 | ts | 🟪 | aalf | ⬛ | lfms | 🟨 |
| imfs-h | 🟩 | imfs-v | 🟦 | | | half | 🟦 | prts | 🟧 | lfms | ⬜ |

## 3.2 MODEL SETUP

We used SPHARM-Net (Ha & Lyu, 2022) for our backbone model. We set the entry channels $C = 128$ and the harmonic bandwidth $L = 80$. We used WGDT signal of $k \in [6, 8, 10]$ and $\sigma = \pi/32$; the optimal value of $\sigma$ for WGDT signal was determined by evaluating performance across multiple configurations as detailed in Appendix A.1. For comparison, we used ADT and Disk with $\sigma \in [\pi/32, 3\pi/64, \pi/16]$, which are small enough to capture fine-grained sulcal branches. We simulated 3 clicks per subject, accumulating the losses from each of the clicks with $\beta_i \in [1/6, 1/3, 1/2]$. The models were trained using Adam optimizer at an initial learning rate 0.01 with decay by a factor of 0.1 if no improvement is made in two consecutive epochs.

## 3.3 TRAINING AND EVALUATION CRITERIA

We simulated user clicks and encoded guidance signals on the unit spherical mesh obtained from the FreeSurfer reconstruction pipeline. For each sulcus in each participant, 10 initial clicks were generated on this unit sphere, where the points were selected to maximize both their distance from the label boundary and mutual separation. Each click served as the initial click location for an individual run, and the performances were averaged across these 10 runs to produce a single performance value per subject. We recorded model performance for up to 3 successive clicks to evaluate the effect of iterative user interaction. This procedure approximates a wider range of user interactions within our limited cohort size.

We re-tessellated individual spheres by icosahedral subdivision of 40,962 vertices (Baumgardner & Frederickson, 1985) before being fed to the model. After the model prediction, the outputs were mapped back to the original sphere and masked out beyond the sulcal regions by keeping only faces that contain at least one vertex with $curv \geq 0$. This masking strategy addresses re-tessellation artifacts and ensures that subsequent user clicks remain within sulcal regions.

We performed 5-fold cross validation: 3 partitions for training and 1 each for validation and test. We used Dice score to evaluate the labeling performance. We conducted paired $t$-tests between WGDT and other encoding schemes. In ROI-wise comparison, multi-comparison correction among the 17 sulci using false discovery rate (FDR) (Benjamini & Hochberg, 1995) at $q = 0.05$ was applied. The experiments were performed on Intel Xeon 6526Y and NVIDIA RTX 6000 Ada Generation.

# 4 RESULTS

## 4.1 SPHERICAL GUIDANCE SIGNALS

We evaluate sulcal labeling performance across different spherical guidance signals, while keeping all other configurations (backbone model, geometric features, etc.) are fixed. The proposed WGDT signal achieves the best performance for a single click compared to ADT and Disk signals as shown in Figure 4. The ROI-wise quantitative results can be found in Appendix A.3. While all encoding schemes perform similarly on large and consistent sulci, the WGDT signal consistently outperforms other signals (adjusted $p < 0.05$) on all of the 9 small and variable sulci, highlighting its strong effectiveness with minimal user input. Qualitative results demonstrate that the WGDT signal can identify the full extent of shallow sulci, whereas other encoding schemes tend to under-segment the region, identifying only the small portions of the sulci (see Figure 6). This is likely because the WGDT signal can adaptively propagate user interactions along local folding patterns.

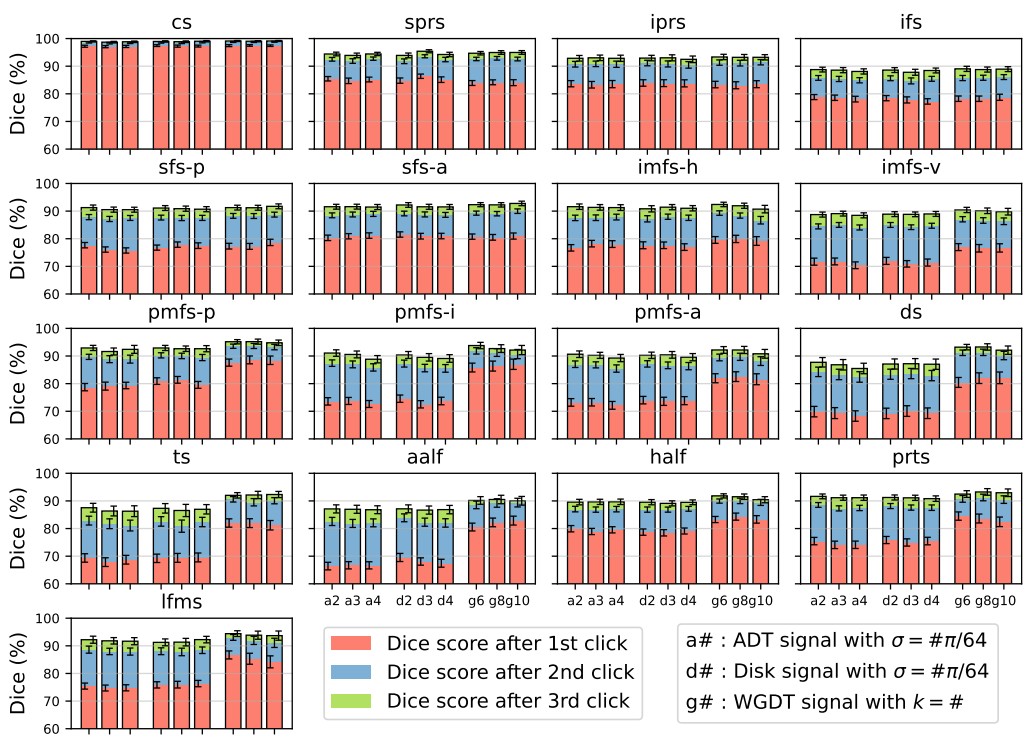

Figure 4: ROI-wise comparison for all guidance signals with iterative refinement. The horizontal axis ticks indicate the guidance signals, shown only on the bottom of each column for better clarity. The vertical axis ticks show the Dice score, whose ranges are shared across all subplots for direct comparison. The WGDT signal shows significant increase in single-click Dice scores compared to the equidistance-based signals in small and variable sulci ($p < 0.05$). Although the performance gap narrows after subsequent clicks in the equidistance-based signals, fewer clicks with the WGDT signal can still reduce human effort.

Specifically, the equidistance-based ADT and Disk signals often fail to sufficiently cover a sulcal region or spillover into adjacent unrelated sulci as shown in Figure 3. Being shape-unaware, these approaches often introduce misattention in unrelated regions during training, potentially leading the model to label sulci overly conservatively or aggressively. In contrast, the WGDT signal allows clicks to limit its influence on shallower sulci along their folding patterns while reducing spillover to adjacent components. This adaptive modulation of click influence based on sulcal patterns helps the model focus on the user's intended refining area. As a result, the ADT or Disk signals require more clicks to refine labels than the WGDT signal.

Due to the relatively small size of variable sulci, the initial click and shape-aware design are critical; see Appendix A.4 for a comparison of their sizes with the largest sulcus. As observed, the WGDT signal with even a single click yields substantial gains over the ADT or Disk signals in small and variable sulci. Although these sulci can be further refined with subsequent clicks regardless of the signal design, they often require considerable user effort. Meanwhile, a large $k$ (i.e., causing the guidance signal to cover a broader area than necessary) can limit the benefit of additional clicks. With a higher $k$, it becomes more difficult to reach statistical significance against the ADT or signals, which results in fewer regions showing better performance than smaller $k$ values. Selecting appropriate $k$ and $\sigma$ values is therefore necessary to balance coverage and precision, which we leave for future work.

It is important to note that none of these methods perfectly follows sulcal regions and their performance can vary depending on the guidance signal size. In this sense, our goal in this study is not to produce signals that exactly cover the regions of interest but rather to provide shape-adaptive signals that minimize spillover; this is commonly observed in simple encoding schemes that ignore under-

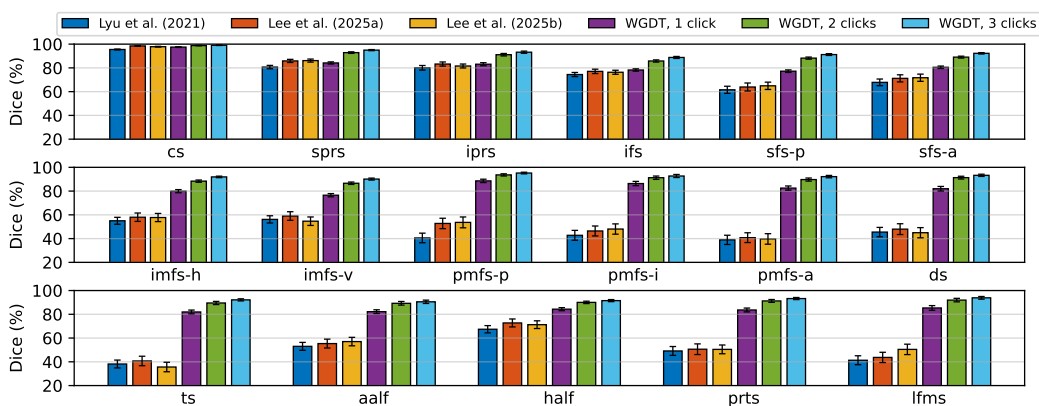

Figure 5: ROI-wise Dice scores of automatic baselines and the proposed interactive model with the WGDT signal ($k = 8$). The WGDT signal significantly outperforms the baselines with a single click in all small sulci, and in large sulci except for cs, sprs, iprs, and ifs ($p < 0.05$). Subsequent clicks significantly improve these four sulci compared to the baselines ($p < 0.05$).

lying neuroanatomy. Although guidance signals are not final labels and may not perfectly align with the manual labels, it is important to minimize their impact on other unrelated regions. Overall, the experimental results suggest that the proposed WGDT signal can help users label shallow sulci with less effort thanks to its shape-aware encoding.

## 4.2 COMPARISON TO AUTOMATIC LABELING METHODS

As no interactive methods are available for sulcal labeling, we instead used the latest fully automatic baselines to evaluate our method: Lyu et al. (2021), Lee et al. (2025a) and Lee et al. (2025b). To the best of our knowledge, these are the only available methods supporting fine-grained sulcal labeling. We used optimal hyperparameter settings reported in their respective publications. Each baseline was retrained on our dataset using the same geometric features from the standard FreeSurfer pipeline (*curv*, *sulc*, and *inflated.H*) to label 17 sulci per subject. By retraining all baselines, we ensure a fair comparison so that observed performance differences reflect the effectiveness of our interactive framework using the WGDT signal. For our method, we used the model the WGDT signal of $k = 8$ over up to 3 user clicks, and the performance of automatic baselines and the proposed framework was compared using Dice scores.

Existing automatic sulcal labeling models often fail to correctly identify small sulci due to their individual anatomical variability, as observed in Figures 5 and 6. The WGDT signal shows significantly higher labeling accuracy than the baselines with a single initial click, achieving improved results in all small sulci and in large sulci except for cs, sprs, iprs and ifs (adjusted $p < 0.05$). An initial click within the target sulcus provides a spatial prior that helps the model resolve ambiguities caused by anatomical variability. Similar to the pattern observed in comparisons across guidance signals, subsequent clicks further refine the labeling. By 2 or 3 clicks, the variable sulci reach near-perfect accuracy. This suggests that incremental user feedback can effectively refine mislabeled regions after the initial labeling. For ROI-wise quantitative results, see Appendix A.5.

## 4.3 RUNTIME ANALYSIS

We evaluated the computational efficiency of our framework by measuring the runtime per single user click. Specifically, we measured the time required for an initial click on the largest sulcus (i.e., central sulcus) across all subjects, each with surfaces containing roughly 100,000 to 170,000 vertices. The same initial click points as in section 3.3 were used. Before measurement, 50 warm-up forward passes were performed to ensure stable performance measurements. Table 2 summarizes the average time for one click, including WGDT signal encoding, re-tessellation from and to icosahedron subdivision, and the model forward pass. On average, a single initial click requires less than 0.5 seconds. This suggests that our framework can provide real-time feedback in practical use.

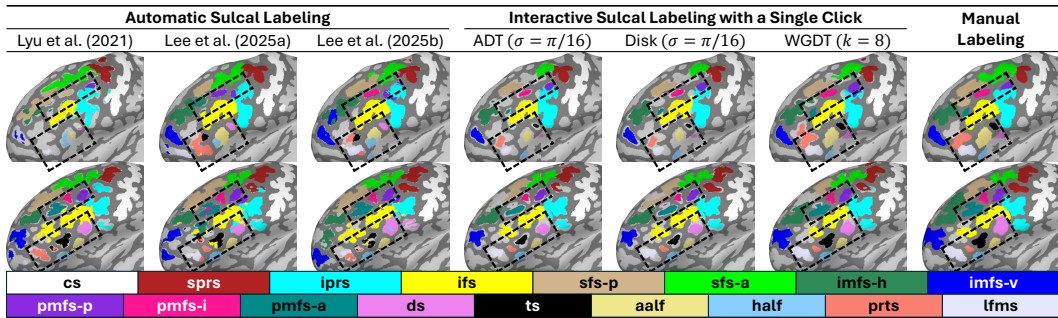

Figure 6: Visual comparison on two example participants across automatic baselines and interactive models with different guidance signals: ADT, Disk ($\sigma = \pi/16$), and WGDT ($k = 8$). WGDT well identifies small and variable sulci in LPFC (highlighted with black dashed boxes), whereas some sulci are not properly identified in automatic baselines and are under-segmented with other signals.

Table 2: Average runtime per initial click on the largest sulcus across all subjects.

| Stage | Runtime (ms) |
| --- | --- |
| WGDT Signal Encoding | $175.47 \pm 24.73$ |
| Re-tessellation | $207.78 \pm 16.77$ |
| Forward Pass | $27.54 \pm 0.05$ |

## 5 DISCUSSION AND CONCLUSION

In this paper, we proposed a shape-adaptive guidance signal for interactive cortical sulcal labeling in LPFC, using the eikonal equation with a curvature-based speed function to account for cortical morphology. The proposed WGDT signal outperforms the equidistance-based signals in small and variable sulci while enabling iterative refinement through multiple clicks.

The comparison with automatic baselines suggests that automatic and interactive frameworks can be jointly used to support the labeling process. The automatic model provides stable predictions for large and consistent sulci, while our proposed interactive model effectively resolves the errors in small and variable sulci with a few clicks. These complementary behaviors indicate that automatic predictions may serve as a useful starting point for our interactive model for further refinement. Therefore, this joint use has the potential to reduce the user interactions in sulcal labeling. These aspects are left for future work.

Despite its promising performance, the proposed approach has several limitations. Although our evaluation was conducted on LPFC with diverse sulcal morphologies, it remains important to consider generalization to other cortical regions. The proposed WGDT signal also involves a relatively wide range of hyperparameters that currently requires manual tuning. Moreover, in cases with excessive noise or pathological anatomy, the proposed curvature-based propagation may be less reliable, and the sulcal labeling task itself may not be well-defined. However, it is noteworthy that this still necessitates accurate surface reconstruction.

Our approach can be further extended by jointly modeling morphologically similar sulci, which may improve generalization to similar types of cortical regions. Also, learning-based strategy could optimize propagation rate of the guidance signal by leveraging geometric information and labeling progression cues. This can reduce the parameter search space for our guidance signal, and improve adaptability to new data. Finally, exploring real-time user-feedback optimization and scalability to full cortical surfaces represents a promising direction for enhancing practical applicability in both research and clinical settings.

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

# A  APPENDIX

## A.1  DETAILED RESULTS FOR WGDT MAXIMUM TRAVEL TIME SELECTION

We performed the experiments to select a suitable $\sigma$ for the WGDT signal. We empirically evaluated performance across multiple sulci and click iterations. As shown in Figure 7, setting $\sigma$ too large tends to degrade performance in small and variable sulci because the guidance signal affects an excessively broad region and reduces precise localization. Table 3 summarizes the detailed Dice scores for all sulci across the varying $\sigma$ and $k$ configurations, showing their mean and standard errors. We chose $\sigma = \pi/32$ for the default parameter setting in the experiments of the main body.

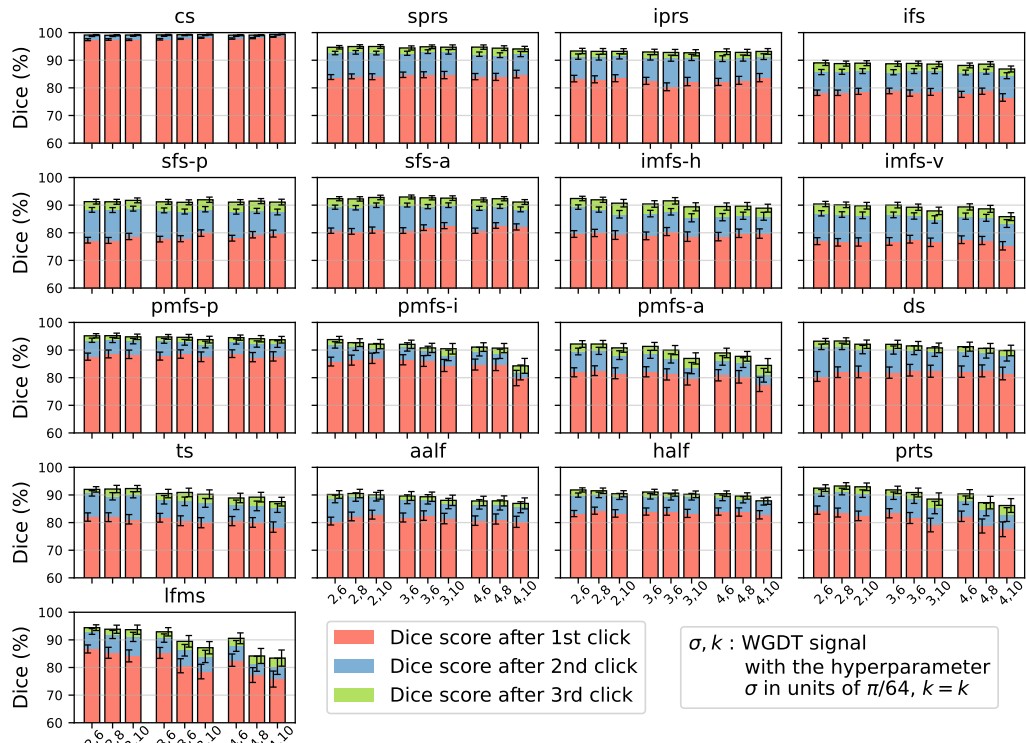

Figure 7: Performance comparison of the WGDT signals with different $\sigma$ values across sulci and click iterations. For small and variable sulci, larger $\sigma$ tends to degrade performance, likely because the guidance signal affects an excessively broad region, reducing precise localization.

Table 3: Performance of the WGDT signals across different $\sigma$ and $k$ with 3 clicks.

| Sulcus | | | cs | | | sprs | | |
|---|---|---|---|---|---|---|---|---|
| Signal | $\sigma$ [$\pi$] | $k$ | Click 1 | Click 2 | Click 3 | Click 1 | Click 2 | Click 3 |
| WGDT | 1/32 | 6 | 97.43±0.35 | 98.77±0.19 | 99.12±0.14 | 83.91±0.88 | 92.61±0.62 | 94.69±0.47 |
| WGDT | 1/32 | 8 | 97.51±0.28 | 98.78±0.21 | 99.03±0.20 | 84.16±0.89 | 92.85±0.65 | 94.95±0.46 |
| WGDT | 1/32 | 10 | 97.42±0.27 | 98.84±0.17 | 99.13±0.15 | 84.03±1.07 | 92.60±0.67 | 94.97±0.49 |
| WGDT | 3/64 | 6 | 97.59±0.27 | 98.84±0.19 | 99.16±0.16 | 84.73±0.98 | 92.52±0.75 | 94.44±0.66 |
| WGDT | 3/64 | 8 | 97.74±0.21 | 99.01±0.12 | 99.23±0.11 | 84.74±1.07 | 93.10±0.67 | 94.87±0.53 |
| WGDT | 3/64 | 10 | 98.14±0.20 | 99.08±0.15 | 99.30±0.14 | 84.63±1.32 | 93.03±0.83 | 94.73±0.65 |
| WGDT | 1/16 | 6 | 97.84±0.23 | 98.85±0.20 | 99.08±0.19 | 84.07±1.13 | 92.19±0.79 | 94.75±0.60 |
| WGDT | 1/16 | 8 | 98.00±0.20 | 98.88±0.19 | 99.08±0.19 | 83.97±1.28 | 91.82±0.86 | 94.37±0.58 |
| WGDT | 1/16 | 10 | 98.46±0.15 | 99.20±0.07 | 99.39±0.05 | 84.99±1.39 | 92.25±0.95 | 94.08±0.76 |

| Sulcus | | | iprs | | | ifs | | |
|--------|--------|----|-------------|-------------|-------------|-------------|-------------|-------------|
| Signal | $\sigma$ [$\pi$] | $k$ | Click 1 | Click 2 | Click 3 | Click 1 | Click 2 | Click 3 |
| WGDT | 1/32 | 6 | 83.35±1.18 | 91.31±0.97 | 93.34±0.82 | 78.23±0.98 | 85.71±0.93 | 89.03±0.82 |
| WGDT | 1/32 | 8 | 83.09±1.27 | 91.10±1.03 | 93.21±0.91 | 78.22±0.98 | 85.81±0.92 | 88.78±0.84 |
| WGDT | 1/32 | 10 | 83.48±1.26 | 91.43±0.96 | 93.22±0.84 | 78.75±1.11 | 86.06±0.89 | 88.93±0.79 |
| WGDT | 3/64 | 6 | 82.53±1.27 | 91.05±0.99 | 92.99±0.85 | 78.89±1.01 | 85.66±0.94 | 88.71±0.86 |
| WGDT | 3/64 | 8 | 80.39±1.42 | 90.74±1.10 | 92.81±0.95 | 78.11±1.13 | 86.06±0.87 | 88.84±0.77 |
| WGDT | 3/64 | 10 | 82.29±1.51 | 91.02±1.12 | 92.71±1.05 | 78.55±1.24 | 86.09±0.94 | 88.60±0.87 |
| WGDT | 1/16 | 6 | 82.15±1.28 | 90.57±1.04 | 93.09±0.90 | 77.65±1.09 | 85.55±0.87 | 88.11±0.78 |
| WGDT | 1/16 | 8 | 82.72±1.44 | 90.78±1.11 | 92.84±0.99 | 78.79±1.12 | 85.87±0.89 | 88.59±0.80 |
| WGDT | 1/16 | 10 | 83.70±1.49 | 91.30±1.04 | 93.18±0.90 | 76.52±1.36 | 84.40±1.08 | 86.83±0.97 |

| Sulcus | | | sfs-p | | | sfs-a | | |
|--------|--------|----|-------------|-------------|-------------|-------------|-------------|-------------|
| Signal | $\sigma$ [$\pi$] | $k$ | Click 1 | Click 2 | Click 3 | Click 1 | Click 2 | Click 3 |
| WGDT | 1/32 | 6 | 77.31±1.03 | 88.27±0.87 | 91.26±0.78 | 80.76±0.96 | 89.26±0.74 | 92.33±0.54 |
| WGDT | 1/32 | 8 | 77.23±1.07 | 88.22±0.89 | 91.21±0.77 | 80.51±1.03 | 89.06±0.79 | 92.26±0.58 |
| WGDT | 1/32 | 10 | 78.67±1.15 | 88.73±0.88 | 91.72±0.75 | 80.97±1.11 | 89.97±0.82 | 92.78±0.63 |
| WGDT | 3/64 | 6 | 77.70±1.02 | 88.10±0.88 | 91.20±0.77 | 80.80±1.02 | 90.03±0.71 | 92.95±0.50 |
| WGDT | 3/64 | 8 | 77.85±1.09 | 87.69±0.90 | 91.03±0.76 | 81.86±1.02 | 89.47±0.76 | 92.65±0.63 |
| WGDT | 3/64 | 10 | 79.88±1.17 | 88.52±0.96 | 91.95±0.75 | 82.59±1.10 | 89.98±0.83 | 92.54±0.74 |
| WGDT | 1/16 | 6 | 78.03±1.04 | 87.66±0.91 | 91.09±0.77 | 80.75±0.94 | 88.83±0.72 | 91.90±0.62 |
| WGDT | 1/16 | 8 | 79.29±1.15 | 87.92±0.92 | 91.46±0.73 | 82.63±1.02 | 89.56±0.76 | 92.32±0.68 |
| WGDT | 1/16 | 10 | 79.70±1.25 | 87.52±1.04 | 91.10±0.86 | 82.11±1.12 | 88.47±0.88 | 91.13±0.79 |

| Sulcus | | | imfs-h | | | imfs-v | | |
|--------|--------|----|-------------|-------------|-------------|-------------|-------------|-------------|
| Signal | $\sigma$ [$\pi$] | $k$ | Click 1 | Click 2 | Click 3 | Click 1 | Click 2 | Click 3 |
| WGDT | 1/32 | 6 | 79.55±1.22 | 89.29±0.81 | 92.40±0.57 | 76.97±1.26 | 87.05±0.95 | 90.41±0.78 |
| WGDT | 1/32 | 8 | 79.93±1.32 | 88.38±0.94 | 91.94±0.63 | 76.54±1.36 | 86.60±1.01 | 90.11±0.82 |
| WGDT | 1/32 | 10 | 79.18±1.58 | 86.70±1.36 | 90.72±1.11 | 76.67±1.50 | 86.32±1.20 | 89.79±0.97 |
| WGDT | 3/64 | 6 | 78.87±1.40 | 86.94±1.18 | 90.42±0.87 | 76.87±1.37 | 86.47±1.03 | 90.01±0.83 |
| WGDT | 3/64 | 8 | 80.44±1.45 | 87.59±1.17 | 91.56±0.84 | 77.68±1.40 | 86.07±1.15 | 89.25±1.00 |
| WGDT | 3/64 | 10 | 78.60±1.72 | 85.42±1.47 | 89.39±1.26 | 76.56±1.52 | 84.80±1.35 | 87.92±1.21 |
| WGDT | 1/16 | 6 | 78.60±1.52 | 85.65±1.32 | 89.50±1.05 | 77.49±1.38 | 86.01±1.13 | 89.36±0.85 |
| WGDT | 1/16 | 8 | 79.82±1.53 | 86.16±1.31 | 89.66±1.04 | 77.21±1.47 | 85.40±1.26 | 88.66±1.07 |
| WGDT | 1/16 | 10 | 79.71±1.69 | 85.98±1.43 | 88.91±1.32 | 75.29±1.51 | 83.35±1.42 | 85.84±1.30 |

| Sulcus | | | pmfs-p | | | pmfs-i | | |
|--------|--------|----|-------------|-------------|-------------|-------------|-------------|-------------|
| Signal | $\sigma$ [$\pi$] | $k$ | Click 1 | Click 2 | Click 3 | Click 1 | Click 2 | Click 3 |
| WGDT | 1/32 | 6 | 87.61±1.29 | 93.54±0.77 | 95.16±0.61 | 85.83±1.60 | 91.86±1.15 | 93.79±0.97 |
| WGDT | 1/32 | 8 | 88.59±1.42 | 93.63±0.92 | 95.20±0.68 | 86.36±1.81 | 91.26±1.40 | 92.69±1.28 |
| WGDT | 1/32 | 10 | 88.46±1.53 | 93.24±0.99 | 94.78±0.78 | 87.01±1.86 | 90.70±1.67 | 92.15±1.47 |
| WGDT | 3/64 | 6 | 87.81±1.44 | 93.34±0.88 | 94.76±0.66 | 86.48±1.83 | 91.13±1.51 | 92.08±1.43 |
| WGDT | 3/64 | 8 | 88.55±1.51 | 92.81±1.04 | 94.62±0.72 | 86.03±1.95 | 89.72±1.67 | 90.78±1.57 |
| WGDT | 3/64 | 10 | 87.56±1.75 | 92.01±1.21 | 93.78±0.92 | 84.42±2.19 | 88.07±1.91 | 90.47±1.54 |
| WGDT | 1/16 | 6 | 88.69±1.39 | 93.10±0.94 | 94.50±0.68 | 84.75±1.98 | 89.42±1.56 | 91.08±1.36 |
| WGDT | 1/16 | 8 | 87.43±1.67 | 92.18±1.15 | 94.10±0.79 | 84.70±2.13 | 88.64±1.68 | 90.71±1.45 |
| WGDT | 1/16 | 10 | 87.70±1.68 | 92.17±1.20 | 93.74±0.95 | 79.88±2.78 | 81.89±2.69 | 84.28±2.29 |

| Sulcus | | | ts | | | aaif | | |
|--------|--------|----|-------------|-------------|-------------|-------------|-------------|-------------|
| Signal | $\sigma$ [$\pi$] | $k$ | Click 1 | Click 2 | Click 3 | Click 1 | Click 2 | Click 3 |
| WGDT | 1/32 | 6 | 82.01±1.51 | 90.59±0.99 | 92.02±0.78 | 80.54±1.42 | 88.56±1.38 | 90.17±1.22 |
| WGDT | 1/32 | 8 | 81.96±1.60 | 89.52±1.33 | 92.15±0.89 | 82.22±1.57 | 89.25±1.53 | 90.51±1.40 |
| WGDT | 1/32 | 10 | 81.20±1.71 | 90.07±1.12 | 92.32±0.90 | 82.88±1.61 | 89.39±1.57 | 90.05±1.54 |
| WGDT | 3/64 | 6 | 81.75±1.58 | 88.26±1.53 | 90.50±0.85 | 81.84±1.63 | 88.44±1.60 | 89.64±1.46 |
| WGDT | 3/64 | 8 | 80.67±1.78 | 87.70±1.54 | 90.91±0.95 | 82.53±1.72 | 88.41±1.69 | 89.36±1.61 |
| WGDT | 3/64 | 10 | 79.95±1.83 | 87.15±1.59 | 90.25±1.08 | 81.46±1.90 | 86.90±1.88 | 88.06±1.74 |
| WGDT | 1/16 | 6 | 80.55±1.66 | 86.43±1.71 | 88.91±1.18 | 80.70±1.71 | 86.60±1.71 | 87.83±1.66 |
| WGDT | 1/16 | 8 | 80.05±1.75 | 86.04±1.74 | 89.21±1.10 | 81.29±1.83 | 86.38±1.83 | 87.84±1.79 |
| WGDT | 1/16 | 10 | 78.37±1.88 | 85.26±1.58 | 87.58±1.31 | 80.27±2.02 | 85.55±1.99 | 86.94±1.97 |

| Sulcus | | | pmfs-a | | | ds | | |
|--------|--------|----|------------|------------|------------|------------|------------|------------|
| Signal | $\sigma$ [$\pi$] | $k$ | Click 1 | Click 2 | Click 3 | Click 1 | Click 2 | Click 3 |
| WGDT | 1/32 | 6 | 81.95±1.64 | 89.47±1.21 | 92.20±0.92 | 80.44±1.79 | 91.17±0.99 | 93.21±0.86 |
| WGDT | 1/32 | 8 | 82.53±1.76 | 89.71±1.29 | 92.18±1.01 | 81.98±1.93 | 91.33±1.16 | 93.28±0.94 |
| WGDT | 1/32 | 10 | 81.59±1.98 | 88.16±1.58 | 90.79±1.34 | 82.10±2.11 | 90.18±1.52 | 92.11±1.19 |
| WGDT | 3/64 | 6 | 82.13±1.84 | 88.45±1.44 | 91.32±1.17 | 81.83±2.03 | 90.00±1.48 | 92.07±1.11 |
| WGDT | 3/64 | 8 | 81.18±2.02 | 86.97±1.66 | 89.98±1.44 | 82.71±2.10 | 89.95±1.60 | 91.51±1.33 |
| WGDT | 3/64 | 10 | 79.54±2.20 | 83.47±2.03 | 86.97±1.73 | 82.34±2.17 | 89.46±1.75 | 90.78±1.59 |
| WGDT | 1/16 | 6 | 80.87±2.04 | 86.23±1.72 | 88.96±1.46 | 82.19±2.05 | 89.33±1.63 | 91.23±1.39 |
| WGDT | 1/16 | 8 | 80.32±2.26 | 85.46±1.77 | 87.73±1.66 | 82.46±2.16 | 88.86±1.76 | 90.63±1.56 |
| WGDT | 1/16 | 10 | 77.57±2.58 | 80.87±2.48 | 84.45±2.22 | 81.51±2.29 | 87.93±1.91 | 89.84±1.79 |

| Sulcus | | | half | | | prts | | |
|--------|--------|----|------------|------------|------------|------------|------------|------------|
| Signal | $\sigma$ [$\pi$] | $k$ | Click 1 | Click 2 | Click 3 | Click 1 | Click 2 | Click 3 |
| WGDT | 1/32 | 6 | 83.27±1.11 | 89.94±0.76 | 91.82±0.57 | 84.51±1.50 | 90.75±1.22 | 92.48±0.88 |
| WGDT | 1/32 | 8 | 84.32±1.28 | 90.02±1.00 | 91.51±0.88 | 83.63±1.62 | 91.24±1.20 | 93.25±0.89 |
| WGDT | 1/32 | 10 | 83.32±1.35 | 89.47±1.10 | 90.43±1.00 | 82.44±1.73 | 90.48±1.37 | 92.96±1.09 |
| WGDT | 3/64 | 6 | 84.00±1.29 | 89.51±0.99 | 91.07±0.77 | 83.38±1.67 | 89.73±1.34 | 91.84±1.10 |
| WGDT | 3/64 | 8 | 84.07±1.39 | 89.28±1.11 | 90.67±0.93 | 81.66±1.99 | 88.54±1.60 | 90.89±1.36 |
| WGDT | 3/64 | 10 | 83.33±1.47 | 89.36±1.20 | 90.24±1.04 | 79.12±2.47 | 85.44±2.23 | 88.51±2.05 |
| WGDT | 1/16 | 6 | 84.00±1.31 | 88.95±1.03 | 90.44±0.83 | 82.28±1.88 | 88.10±1.54 | 90.38±1.30 |
| WGDT | 1/16 | 8 | 83.83±1.50 | 88.28±1.15 | 89.63±0.94 | 78.77±2.54 | 84.74±2.28 | 87.23±2.09 |
| WGDT | 1/16 | 10 | 82.82±1.50 | 87.62±1.27 | 87.88±1.16 | 77.58±2.66 | 82.90±2.50 | 86.18±2.32 |

| Sulcus | | | lfms | | |
|--------|--------|----|------------|------------|------------|
| Signal | $\sigma$ [$\pi$] | $k$ | Click 1 | Click 2 | Click 3 |
| WGDT | 1/32 | 6 | 86.71±1.45 | 93.05±1.05 | 94.41±0.81 |
| WGDT | 1/32 | 8 | 85.33±1.99 | 91.95±1.47 | 93.86±1.13 |
| WGDT | 1/32 | 10 | 84.20±2.17 | 91.08±1.67 | 93.69±1.25 |
| WGDT | 3/64 | 6 | 85.27±1.97 | 90.65±1.51 | 92.92±1.27 |
| WGDT | 3/64 | 8 | 80.63±2.55 | 86.24±2.09 | 89.48±1.79 |
| WGDT | 3/64 | 10 | 78.53±2.64 | 84.01±2.23 | 87.16±2.20 |
| WGDT | 1/16 | 6 | 82.66±2.23 | 87.80±1.96 | 90.56±1.74 |
| WGDT | 1/16 | 8 | 77.27±2.68 | 81.36±2.72 | 84.18±2.35 |
| WGDT | 1/16 | 10 | 75.81±2.90 | 80.42±2.93 | 83.41±2.57 |

## A.2 DETAILED LPFC SULCAL DEFINITIONS

Following the most recent LPFC sulcal definitions (Petrides, 2018; Voorhies et al., 2021; Willbrand et al., 2022; Yao et al., 2023; Willbrand et al., 2023; 2024), we manually defined LPFC sulci on the left hemisphere for the 72 participants from the HCP. The 8 large sulci included the central sulcus (cs), superior (sprs) and inferior (iprs) components of the precentral sulcus, which forms the posterior boundary of LPFC, as well as the inferior frontal sulcus (ifs), which extends longitudinally across LPFC. Anterior to precentral sulcus and ifs are anterior (sfs-a) and posterior (sfs-p) components of the superior frontal sulcus, and the horizontal (imfs-h) and vertical (imfs-v) components of the intermediate frontal sulcus. These sulci are deep and consistent across participants.

The 9 smaller sulci included the posterior (pmfs-p), intermediate (pmfs-i), and anterior (pmfs-a) portions of the middle frontal sulcus, which are located superior to the ifs, as well as the diagonal sulcus (ds), triangular sulcus (ts), pretriangular sulcus (prts), the lateral frontomarginal sulcus (lfms), and the ascending (aalf) and horizontal (half) rami of the lateral fissure, located inferior to the ifs. These sulci were shallower and more variable across participants. Interested readers are referred to (Petrides, 2018; Voorhies et al., 2021; Willbrand et al., 2022; Yao et al., 2023; Willbrand et al., 2023; 2024) for detailed discussions of their characteristics.

## A.3 DETAILED RESULTS FOR GUIDANCE SIGNAL COMPARISON

In this section, we provide the detailed numerical results underlying the performance comparison among guidance signals (ADT, Disk, and WGDT) for sulcal labeling with 3 clicks. The results provided in Table 4 complement the summary presented in the main text (Figure 4), showing the mean and standard error of Dice scores for each sulcus and click iteration.

Table 4: Performance comparison among ADT, Disk, and WGDT on sulcal labeling with 3 clicks.

| Sulcus | | | cs | | | sprs | | |
|--------|--------|----|---------|---------|---------|---------|---------|---------|
| Signal | $\sigma$ [$\pi$] | $k$ | Click 1 | Click 2 | Click 3 | Click 1 | Click 2 | Click 3 |
| ADT | 1/32 | – | 97.22±0.36 | 98.60±0.22 | 98.97±0.18 | 85.45±0.80 | 92.49±0.67 | 94.41±0.59 |
| ADT | 3/64 | – | 97.06±0.38 | 98.34±0.29 | 98.70±0.24 | 84.70±0.99 | 91.87±0.83 | 93.91±0.72 |
| ADT | 1/16 | – | 97.14±0.39 | 98.45±0.27 | 98.83±0.22 | 85.18±0.86 | 92.73±0.60 | 94.43±0.50 |
| Disk | 1/32 | – | 97.44±0.32 | 98.59±0.24 | 98.95±0.20 | 84.78±0.95 | 91.78±0.86 | 93.90±0.77 |
| Disk | 3/64 | – | 97.26±0.37 | 98.47±0.29 | 98.86±0.23 | 86.36±0.79 | 93.58±0.56 | 95.41±0.46 |
| Disk | 1/16 | – | 97.27±0.34 | 98.67±0.25 | 99.03±0.20 | 85.12±1.01 | 92.41±0.80 | 94.24±0.73 |
| WGDT | 1/32 | 6 | 97.43±0.35 | 98.77±0.19 | 99.12±0.14 | 83.91±0.88 | 92.61±0.62 | 94.69±0.47 |
| WGDT | 1/32 | 8 | 97.51±0.28 | 98.78±0.21 | 99.03±0.20 | 84.16±0.89 | 92.85±0.65 | 94.95±0.46 |
| WGDT | 1/32 | 10 | 97.42±0.27 | 98.84±0.17 | 99.13±0.15 | 84.03±1.07 | 92.60±0.67 | 94.97±0.49 |

| Sulcus | | | iprs | | | ifs | | |
|--------|--------|----|---------|---------|---------|---------|---------|---------|
| Signal | $\sigma$ [$\pi$] | $k$ | Click 1 | Click 2 | Click 3 | Click 1 | Click 2 | Click 3 |
| ADT | 1/32 | – | 83.66±1.12 | 90.71±1.02 | 92.86±0.94 | 78.82±0.92 | 85.75±0.86 | 88.77±0.85 |
| ADT | 3/64 | – | 83.28±1.19 | 90.85±1.00 | 93.02±0.89 | 78.57±0.97 | 85.38±0.94 | 88.57±0.85 |
| ADT | 1/16 | – | 83.51±1.30 | 90.71±1.13 | 92.86±0.99 | 78.08±1.03 | 84.91±0.94 | 88.10±0.86 |
| Disk | 1/32 | – | 83.92±1.17 | 91.13±1.00 | 92.94±0.93 | 78.43±0.94 | 85.63±0.88 | 88.59±0.80 |
| Disk | 3/64 | – | 83.86±1.24 | 90.82±1.10 | 93.03±0.95 | 77.79±1.11 | 84.67±1.08 | 87.80±1.03 |
| Disk | 1/16 | – | 83.78±1.28 | 90.47±1.19 | 92.49±1.04 | 77.25±0.98 | 85.43±0.88 | 88.44±0.77 |
| WGDT | 1/32 | 6 | 83.35±1.18 | 91.31±0.97 | 93.34±0.82 | 78.23±0.98 | 85.71±0.93 | 89.03±0.82 |
| WGDT | 1/32 | 8 | 83.09±1.27 | 91.10±1.03 | 93.21±0.91 | 78.22±0.98 | 85.81±0.92 | 88.78±0.84 |
| WGDT | 1/32 | 10 | 83.48±1.26 | 91.43±0.96 | 93.22±0.84 | 78.75±1.11 | 86.06±0.89 | 88.93±0.79 |

| Sulcus | | | sfs-p | | | sfs-a | | |
|--------|--------|----|---------|---------|---------|---------|---------|---------|
| Signal | $\sigma$ [$\pi$] | $k$ | Click 1 | Click 2 | Click 3 | Click 1 | Click 2 | Click 3 |
| ADT | 1/32 | – | 77.66±0.97 | 87.82±0.93 | 91.27±0.87 | 80.32±0.99 | 88.47±0.88 | 91.58±0.72 |
| ADT | 3/64 | – | 76.09±0.96 | 87.15±0.93 | 90.53±0.85 | 80.90±0.96 | 88.77±0.88 | 91.61±0.73 |
| ADT | 1/16 | – | 75.75±0.95 | 87.52±0.90 | 90.51±0.88 | 81.14±1.00 | 88.99±0.92 | 91.47±0.84 |
| Disk | 1/32 | – | 76.79±0.92 | 87.65±0.88 | 91.11±0.75 | 81.53±0.96 | 89.13±0.85 | 92.22±0.73 |
| Disk | 3/64 | – | 77.86±0.96 | 87.50±0.94 | 90.88±0.89 | 80.92±1.03 | 88.80±0.90 | 91.56±0.80 |
| Disk | 1/16 | – | 77.50±1.02 | 87.54±0.92 | 90.69±0.86 | 80.97±0.99 | 88.63±0.87 | 91.52±0.74 |
| WGDT | 1/32 | 6 | 77.31±1.03 | 88.27±0.87 | 91.26±0.78 | 80.76±0.96 | 89.26±0.74 | 92.33±0.54 |
| WGDT | 1/32 | 8 | 77.23±1.07 | 88.22±0.89 | 91.21±0.77 | 80.51±1.03 | 89.06±0.79 | 92.26±0.58 |
| WGDT | 1/32 | 10 | 78.67±1.15 | 88.73±0.88 | 91.72±0.75 | 80.97±1.11 | 89.97±0.82 | 92.78±0.63 |

| Sulcus | | | imfs-h | | | imfs-v | | |
|--------|--------|----|---------|---------|---------|---------|---------|---------|
| Signal | $\sigma$ [$\pi$] | $k$ | Click 1 | Click 2 | Click 3 | Click 1 | Click 2 | Click 3 |
| ADT | 1/32 | – | 76.70±1.16 | 87.54±0.98 | 91.60±0.76 | 71.70±1.27 | 84.46±0.91 | 88.74±0.74 |
| ADT | 3/64 | – | 78.22±1.18 | 87.60±1.02 | 91.35±0.78 | 71.77±1.24 | 84.99±0.87 | 89.09±0.72 |
| ADT | 1/16 | – | 78.06±1.22 | 87.84±1.04 | 91.27±0.83 | 70.39±1.22 | 84.07±0.91 | 88.46±0.77 |
| Disk | 1/32 | – | 77.63±1.25 | 87.21±1.13 | 90.83±0.95 | 71.97±1.27 | 84.98±0.88 | 88.89±0.73 |
| Disk | 3/64 | – | 77.62±1.19 | 87.97±0.95 | 91.42±0.76 | 70.91±1.17 | 84.17±0.85 | 88.83±0.70 |
| Disk | 1/16 | – | 76.97±1.21 | 87.58±0.96 | 91.03±0.79 | 71.37±1.28 | 84.70±0.90 | 88.97±0.75 |
| WGDT | 1/32 | 6 | 79.55±1.22 | 89.29±0.81 | 92.40±0.57 | 76.97±1.26 | 87.05±0.95 | 90.41±0.78 |
| WGDT | 1/32 | 8 | 79.93±1.32 | 88.38±0.94 | 91.94±0.63 | 76.54±1.36 | 86.60±1.01 | 90.11±0.82 |
| WGDT | 1/32 | 10 | 79.18±1.58 | 86.70±1.36 | 90.72±1.11 | 76.67±1.50 | 86.32±1.20 | 89.79±0.97 |

| Sulcus | | | pmfs-p | | | pmfs-i | | |
|---|---|---|---|---|---|---|---|---|
| Signal | $\sigma$ [$\pi$] | $k$ | Click 1 | Click 2 | Click 3 | Click 1 | Click 2 | Click 3 |
| ADT | 1/32 | – | 78.73±1.32 | 89.70±0.94 | 92.92±0.79 | 73.58±1.34 | 87.42±1.18 | 91.07±1.16 |
| ADT | 3/64 | – | 79.10±1.45 | 88.85±1.26 | 91.63±1.27 | 73.73±1.27 | 86.97±1.24 | 90.57±1.30 |
| ADT | 1/16 | – | 79.33±1.18 | 88.87±1.44 | 92.41±1.00 | 72.63±1.26 | 85.88±1.32 | 88.84±1.67 |
| Disk | 1/32 | – | 80.92±1.20 | 90.24±0.93 | 92.91±0.89 | 74.55±1.33 | 87.25±1.25 | 90.37±1.32 |
| Disk | 3/64 | – | 81.38±1.22 | 90.03±0.99 | 92.64±0.90 | 72.52±1.32 | 85.63±1.35 | 89.52±1.59 |
| Disk | 1/16 | – | 79.55±1.25 | 89.44±1.12 | 92.65±1.05 | 73.72±1.36 | 85.52±1.38 | 89.11±1.44 |
| WGDT | 1/32 | 6 | 87.61±1.29 | 93.54±0.77 | 95.16±0.61 | 85.83±1.60 | 91.86±1.15 | 93.79±0.97 |
| WGDT | 1/32 | 8 | 88.59±1.42 | 93.63±0.92 | 95.20±0.68 | 86.36±1.81 | 91.26±1.40 | 92.69±1.28 |
| WGDT | 1/32 | 10 | 88.46±1.53 | 93.24±0.99 | 94.78±0.78 | 87.01±1.86 | 90.70±1.67 | 92.15±1.47 |

| Sulcus | | | pmfs-a | | | ds | | |
|---|---|---|---|---|---|---|---|---|
| Signal | $\sigma$ [$\pi$] | $k$ | Click 1 | Click 2 | Click 3 | Click 1 | Click 2 | Click 3 |
| ADT | 1/32 | – | 73.21±1.37 | 87.02±1.18 | 90.65±1.29 | 69.87±1.90 | 84.26±1.71 | 87.73±1.68 |
| ADT | 3/64 | – | 73.26±1.35 | 86.79±1.10 | 90.24±1.18 | 69.34±1.99 | 83.33±1.90 | 86.83±1.90 |
| ADT | 1/16 | – | 72.14±1.40 | 85.46±1.31 | 89.26±1.34 | 68.27±1.89 | 82.42±1.90 | 85.49±2.14 |
| Disk | 1/32 | – | 74.02±1.36 | 87.15±1.15 | 90.25±1.22 | 69.23±1.89 | 83.41±1.84 | 87.11±1.85 |
| Disk | 3/64 | – | 73.60±1.47 | 86.63±1.21 | 90.42±1.25 | 70.08±1.93 | 83.65±1.85 | 87.21±1.65 |
| Disk | 1/16 | – | 73.85±1.46 | 86.33±1.31 | 89.57±1.46 | 69.29±1.92 | 82.91±1.86 | 87.03±1.69 |
| WGDT | 1/32 | 6 | 81.95±1.64 | 89.47±1.21 | 92.20±0.92 | 80.44±1.79 | 91.17±0.99 | 93.21±0.86 |
| WGDT | 1/32 | 8 | 82.53±1.76 | 89.71±1.29 | 92.18±1.01 | 81.98±1.93 | 91.33±1.16 | 93.28±0.94 |
| WGDT | 1/32 | 10 | 81.59±1.98 | 88.16±1.58 | 90.79±1.34 | 82.10±2.11 | 90.18±1.52 | 92.11±1.19 |

| Sulcus | | | ts | | | aaif | | |
|---|---|---|---|---|---|---|---|---|
| Signal | $\sigma$ [$\pi$] | $k$ | Click 1 | Click 2 | Click 3 | Click 1 | Click 2 | Click 3 |
| ADT | 1/32 | – | 69.36±1.53 | 82.87±1.59 | 87.54±1.36 | 66.41±1.38 | 82.57±1.47 | 87.11±1.38 |
| ADT | 3/64 | – | 67.85±1.57 | 81.74±1.78 | 86.35±1.84 | 66.71±1.32 | 81.85±1.45 | 86.97±1.33 |
| ADT | 1/16 | – | 68.72±1.58 | 81.09±1.96 | 86.28±1.75 | 66.68±1.30 | 82.06±1.43 | 86.81±1.41 |
| Disk | 1/32 | – | 69.23±1.52 | 82.58±1.72 | 87.32±1.61 | 69.53±1.47 | 83.89±1.43 | 87.14±1.44 |
| Disk | 3/64 | – | 69.39±1.61 | 80.87±2.21 | 86.54±1.70 | 68.07±1.33 | 82.31±1.47 | 86.68±1.44 |
| Disk | 1/16 | – | 69.54±1.61 | 82.36±1.70 | 86.97±1.62 | 67.45±1.43 | 82.03±1.52 | 86.85±1.34 |
| WGDT | 1/32 | 6 | 82.01±1.51 | 90.59±0.99 | 92.02±0.78 | 80.54±1.42 | 88.56±1.38 | 90.17±1.22 |
| WGDT | 1/32 | 8 | 81.96±1.60 | 89.52±1.33 | 92.15±0.89 | 82.22±1.57 | 89.25±1.53 | 90.51±1.40 |
| WGDT | 1/32 | 10 | 81.20±1.71 | 90.07±1.12 | 92.32±0.90 | 82.88±1.61 | 89.39±1.57 | 90.05±1.54 |

| Sulcus | | | half | | | prts | | |
|---|---|---|---|---|---|---|---|---|
| Signal | $\sigma$ [$\pi$] | $k$ | Click 1 | Click 2 | Click 3 | Click 1 | Click 2 | Click 3 |
| ADT | 1/32 | – | 79.94±1.15 | 87.04±1.00 | 89.58±0.83 | 75.46±1.34 | 88.53±0.87 | 91.64±0.71 |
| ADT | 3/64 | – | 78.93±1.12 | 86.86±0.95 | 89.65±0.79 | 74.10±1.35 | 87.30±0.93 | 91.16±0.74 |
| ADT | 1/16 | – | 79.54±1.12 | 87.02±0.99 | 89.68±0.75 | 74.12±1.34 | 87.45±0.97 | 91.17±0.82 |
| Disk | 1/32 | – | 78.71±1.14 | 86.86±0.95 | 89.54±0.79 | 75.85±1.29 | 88.14±0.91 | 91.24±0.73 |
| Disk | 3/64 | – | 78.46±1.15 | 86.85±0.92 | 89.12±0.72 | 74.96±1.31 | 87.59±0.94 | 91.13±0.73 |
| Disk | 1/16 | – | 79.11±1.09 | 87.14±0.87 | 89.50±0.69 | 75.47±1.36 | 87.55±1.00 | 90.83±0.78 |
| WGDT | 1/32 | 6 | 83.27±1.11 | 89.94±0.76 | 91.82±0.57 | 84.51±1.50 | 90.75±1.22 | 92.48±0.88 |
| WGDT | 1/32 | 8 | 84.32±1.28 | 90.02±1.00 | 91.51±0.88 | 83.63±1.62 | 91.24±1.20 | 93.25±0.89 |
| WGDT | 1/32 | 10 | 83.32±1.35 | 89.47±1.10 | 90.43±1.00 | 82.44±1.73 | 90.48±1.37 | 92.96±1.09 |

| Sulcus | | | lfms | | |
|---|---|---|---|---|---|
| Signal | $\sigma$ [$\pi$] | $k$ | Click 1 | Click 2 | Click 3 |
| ADT | 1/32 | – | 75.43±1.10 | 88.62±1.28 | 92.22±1.34 |
| ADT | 3/64 | – | 74.73±1.07 | 87.97±1.17 | 91.82±1.13 |
| ADT | 1/16 | – | 74.81±1.10 | 87.93±1.28 | 91.68±1.35 |
| Disk | 1/32 | – | 75.87±1.13 | 88.04±1.24 | 91.26±1.32 |
| Disk | 3/64 | – | 75.95±1.21 | 87.79±1.37 | 91.34±1.40 |
| Disk | 1/16 | – | 76.34±1.15 | 88.40±1.18 | 92.25±1.09 |
| WGDT | 1/32 | 6 | 86.71±1.45 | 93.05±1.05 | 94.41±0.81 |
| WGDT | 1/32 | 8 | 85.33±1.99 | 91.95±1.47 | 93.86±1.13 |
| WGDT | 1/32 | 10 | 84.20±2.17 | 91.08±1.67 | 93.69±1.25 |

## A.4 Average Area of LPFC Sulci

Figure 8 shows the average area ratio of each sulcus relative to the central sulcus (the largest sulcus included in this study). The comparison highlights that variable sulci occupy only a small portion of the cortex compared to large sulci. As a result, subsequent refinements for these sulci are less reliable and more difficult for users to perform accurately. Hence, achieving high accuracy from the initial interaction is particularly important.

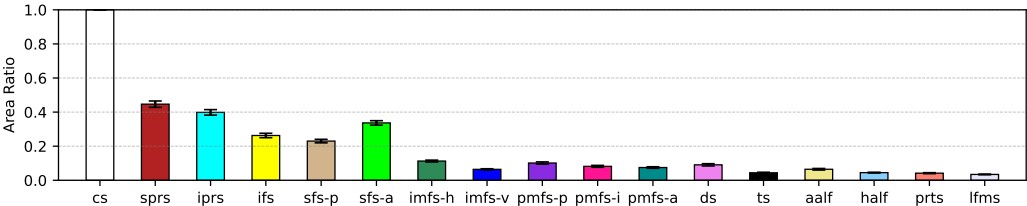

Figure 8: Average area ratio between central sulcus (cs) and the other sulci defined on LPFC. The variable sulci are much smaller compared to the central sulcus.

## A.5 Detailed Results for Comparison to Automatic Labeling Methods

In this section, We report the detailed numerical results underlying the performance comparison among automatic baselines of Lyu et al. (2021), Lee et al. (2025a), Lee et al. (2025b) and the proposed interactive model using the WGDT signal ($k = 8$) up to 3 clicks. The results provided in Table 5 complement the summary presented in the main text (Figure 5), showing the mean and standard error of Dice scores for each sulcus and click iteration.

Table 5: Performance comparison among non-interactive baselines and WGDT up to 3 clicks.

| Method | cs | sprs | iprs | ifs | sfs-p | sfs-a |
|---|---|---|---|---|---|---|
| Lyu et al. (2021) | 95.49±0.36 | 80.68±1.41 | 80.08±1.93 | 74.41±1.73 | 61.55±2.99 | 67.81±2.81 |
| Lee et al. (2025a) | 98.49±0.29 | 85.86±1.32 | 83.23±1.74 | 77.07±1.83 | 63.91±3.37 | 71.18±3.00 |
| Lee et al. (2025b) | 97.83±0.31 | 86.14±1.29 | 81.59±1.65 | 76.32±1.69 | 64.91±3.07 | 71.71±3.03 |
| WGDT, 1 click | 97.51±0.28 | 84.16±0.89 | 83.09±1.27 | 78.22±0.98 | 77.23±1.07 | 80.51±1.03 |
| WGDT, 2 clicks | 98.78±0.21 | 92.85±0.65 | 91.10±1.03 | 85.81±0.92 | 88.22±0.89 | 89.06±0.79 |
| WGDT, 3 clicks | 99.03±0.20 | 94.95±0.46 | 93.21±0.91 | 88.78±0.84 | 91.21±0.77 | 92.26±0.58 |

| Method | imfs-h | imfs-v | pmfs-p | pmfs-i | pmfs-a | ds |
|---|---|---|---|---|---|---|
| Lyu et al. (2021) | 55.00±2.91 | 56.18±3.04 | 40.55±4.03 | 42.78±4.16 | 38.95±3.89 | 45.50±3.93 |
| Lee et al. (2025a) | 58.01±3.53 | 59.00±3.62 | 52.77±4.37 | 46.43±4.18 | 40.81±4.12 | 47.97±4.54 |
| Lee et al. (2025b) | 57.75±3.43 | 54.66±3.57 | 53.65±4.56 | 48.06±4.33 | 39.72±4.43 | 44.97±4.28 |
| WGDT, 1 click | 79.93±1.32 | 76.54±1.36 | 88.59±1.42 | 86.36±1.81 | 82.53±1.76 | 81.98±1.93 |
| WGDT, 2 clicks | 88.38±0.94 | 86.60±1.01 | 93.63±0.92 | 91.26±1.40 | 89.71±1.29 | 91.33±1.16 |
| WGDT, 3 clicks | 91.94±0.63 | 90.11±0.82 | 95.20±0.68 | 92.69±1.28 | 92.18±1.01 | 93.28±0.94 |

| Method | ts | aaif | half | prts | lfms |
|---|---|---|---|---|---|
| Lyu et al. (2021) | 38.15±3.31 | 53.02±3.32 | 67.42±3.06 | 49.14±3.73 | 41.33±3.77 |
| Lee et al. (2025a) | 40.71±3.99 | 55.35±3.72 | 72.71±3.35 | 50.61±4.49 | 43.70±4.32 |
| Lee et al. (2025b) | 35.62±4.00 | 57.04±3.57 | 71.26±3.25 | 50.47±3.69 | 50.46±4.37 |
| WGDT, 1 click | 81.96±1.60 | 82.22±1.57 | 84.32±1.28 | 83.63±1.62 | 85.33±1.99 |
| WGDT, 2 clicks | 89.52±1.33 | 89.25±1.53 | 90.02±1.00 | 91.24±1.20 | 91.95±1.47 |
| WGDT, 3 clicks | 92.15±0.89 | 90.51±1.40 | 91.51±0.88 | 93.25±0.89 | 93.86±1.13 |

