# OpenReview forum: "Shape-Adaptive Guidance Signal for Interactive Cortical Sulcal Labeling"
_ICLR.cc/2026/Conference — Submitted to ICLR 2026_

### Official Review · Reviewer_njFB · 2025-10-30

**Soundness:** 3
**Presentation:** 3
**Contribution:** 3
**Rating:** 6
**Confidence:** 5

**Summary:**

This paper focused on a shape-adaptive guidance signal for promptable segmentation of cortical sulci. The authors proposed a curvature-aware encoding of user clicks that can provide the underlying cortical geometry. The results show improving labeling accuracy with minimal user prompts. The proposed method highlights the spherical representation; the surfaces are mapped to the unit sphere with neuroimaging pipelines. Then it includes an interactive framework and a shape-adaptive guidance signal.

**Strengths:**

- The design of the simulated guidance signal for prompt segmentation is new and attractive for sulci analysis.
- The promptable segmentation for cortical surface segmentation is critical as interactive analysis for subregions are demanding in the domain.
- The results show improved accuracy for tiny structures in the challenging sulci.

**Weaknesses:**

- The design of curvation estimation can be more clarified. Such as the curvature estimation on discrete meshes can be noisy in shallow sulci. The errors could affect the propagation speed function and mislead the signal.
- The segmentation can only provide prompt-based binary segmentation right? No multi-label, semantic differentiations?
- In limited regions, the method could be more validated beyond the LPFC patterns.
- The real cortical surface segmentation data is hard to get, but the studies used in this work is relatively too small. In the future, a larger cohort evaluation might be needed.

**Questions:**

Questions and suggestions are associated with the weakness section. Thanks.

---

> ### Author Response · Authors · 2025-11-27
>
> Thank you very much for your careful review and valuable comments. We have addressed your concerns in our common response. For your convenience, we provide a summarized version of the responses to your specific points below. For more detailed explanations, we kindly refer you to the official comment.
>
> **Sensitivity to Curvature Noise & Atypical Brains**
>
> Thank you for pointing out the potential issues with curvature estimation. Our experiments are based on **well-reconstructed cortical surfaces** from healthy participants. Excessively noisy reconstructions may affect the propagation speed function. Although the current WGDT does not explicitly handle such extreme cases, it supports a wide variety of small and variable sulcal morphologies. This enables accurate labeling with only a few clicks per sulcus.
>
> **Clarifications in Interactive Framework & Generalization beyond the LPFC Sulci**
>
> As you have pointed out, multi-label or semantic differentiation is not addressed in current study. Extending single-instance interactive segmentation to multiple objects is non-trivial and introduces excessive complexities. We focused on training **per-sulcus models separately**, allowing each model to specialize in the characteristics of individual sulci.
>
> We trained and evaluated our framework on the LPFC because it contains **a wide variety of sulcal morphologies**. It allows us to demonstrate how well the guidance signal handles different sulcal patterns. While evaluation in other cortical regions is important, the LPFC enables a structured assessment of the guidance signal across diverse sulcal morphologies.
>
> **Limited Cohort Size**
>
> Thank you for your concern regarding our relatively small dataset size (**N=72**). As discussed in the introduction, the acquistion of manual sulcal labels is a labor-intensive task for the trained neuroanatomists. Due to this infeasibility of large-scale data acquisition, the small dataset size may introduce a bias in evaluation as some clicks could favor certain methods.To mitigate such potential bias, we have already generated **diverse initial clicks** for each sulcus, as described in Section 3.3. Specifically, we sampled the click points to maximize their distance from the label boundary and from each other. This strategy approximates a wider range of interaction patterns. It also enables systematic and reproducible evaluation within the limited dataset. We have revised the corresponding section (Section 3.3) for detailed description of this procedure.

---

### Official Review · Reviewer_XRiF · 2025-10-31

**Soundness:** 2
**Presentation:** 2
**Contribution:** 3
**Rating:** 4
**Confidence:** 4

**Summary:**

This paper introduces a novel shape-adaptive guidance signal for interactive segmentation of cortical sulci, particularly focusing on small and shallow sulci in the lateral prefrontal cortex. The authors leverage spherical convolutional neural networks and propose a curvature-aware encoding scheme based on the eikonal equation, which incorporates mean curvature to guide segmentation more effectively than traditional equidistance-based methods. The method is validated on a dataset of 72 subjects with 17 manually labeled sulci, showing that even a single user click using the proposed guidance signal significantly improves segmentation accuracy compared to fully automatic methods and simpler interactive schemes.

**Strengths:**

1. The paper is well-motivated by the challenges in labeling shallow sulci and the limitations of automatic methods.

2. The method requires minimal user input and shows promise for reducing annotation effort in large-scale studies.

3. The paper provides a thorough mathematical formulation of the guidance signal encoding.

4. The use of mean curvature in the eikonal equation to encode user clicks is a novel approach that adapts to the anatomical structure of cortical folds.

**Weaknesses:**

1. The method is evaluated only on LPFC sulci, and its generalizability to other cortical regions is not explored. Moreover, the segmentation is binary per sulcus, which may be inefficient when labeling multiple sulci simultaneously.

2. Although the method is interactive, its performance likely depends on the quality and location of user clicks. The paper does not report real-world user studies to validate the simulated click strategy.

3. The performance of WGDT is sensitive to parameters such as $\sigma$ and $k$, requiring empirical tuning to select optimal values. This may limit out-of-the-box usability for new datasets.

4. While visual comparisons are included, the paper lacks discussion of user interface design and real-time feedback, both of which are important for practical deployment.

**Questions:**

1. The method is evaluated only on LPFC sulci. Have you tested it on other small or shallow sulci out of the box? Since the segmentation is binary per sulcus, this should be conceptually straightforward, similar to SAM.

2. Why were only fully automatic methods used as baselines? Could other interactive segmentation approaches—such as SAM-based projections—be adapted for comparison?

3. Were the automatic baselines retrained or fine-tuned on the same dataset, or were they used as-is?

4. The paper mentions tuning $\sigma$ and $k$ for WGDT. Is there a principled way to select these parameters, or is tuning entirely empirical?

5. How sensitive is the WGDT signal to inaccuracies in curvature estimation? Could noise or bias in curvature maps degrade propagation quality?

6. Is the curvature-based speed function empirically justified beyond visual inspection? Have you compared it with other geometric cues, such as sulcal depth or convexity?

7. Is the model capable of providing real-time feedback during annotation yet? If not, how close is it to achieving this? Please specify the latency and computational requirements per user click.

8. Is there a point of diminishing returns in accuracy after a certain number of clicks? How does performance after saturation compare to expert annotations?

9. How does the model perform on atypical brains (e.g., pediatric or pathological cases)? Is the curvature-based propagation robust to such anatomical variability?

---

> ### Author Response · Authors · 2025-11-27
>
> Thank you very much for the thoughtful and constructive review. We sincerely appreciate the time and effort you invested in evaluating our work. Some of your questions overlap with points raised by other reviewers and are addressed in our common response. We respond below to the remaining points specific to your review.
>
> **WGDT Parameter Selection & Sensitivity**
>
> Regarding the parameter selection for the WGDT signal, the maximum travel time parameter $\sigma$ was chosen empirically based on experiments to capture fine-grained sulcal features without overextending propagation (see Appendix A.1 for details on the experimental selection process and rationale). For the propagation speed modulation parameter $k$, we present multiple configurations in the main text to illustrate its effect on performance.
>
> Across these settings, the proposed guidance signal consistently outperforms the baseline signals. Nevertheless, we acknowledge that the current WGDT signal has a **relatively large parameter search space** and requires some manual tuning. As a next step, learning-based strategies could optimize propagation distances automatically by **leveraging geometric information and labeling progression cues**. This would reduce the need for empirical tuning and improve applicability to new datasets. We have revised the discussion section (Section 5) to address this point.
>
> **Justification of Curvature-Based Speed Function**
>
> We selected **mean curvature** to design the propagation speed function of our guidance signal, as it effectively captures **fine-grained sulcal patterns**. While other geometric and anatomic cues such as average convexity can represent coarser cortical structure, they do not adequately capture the detailed morphology of small sulci. Mean curvature therefore provides a reliable way to constrain propagation within the intended sulcal regions.
>
> **Accuracy Saturation with Additional Clicks**
>
> We intentionally limited interactions to **three clicks per sulcus.** Large and consistent sulci are **generally well-identified** with only a few clicks. Small and variable sulci **may require additional clicks** for accurate labeling. Nevertheless, performing an **excessive number of clicks on these sulci could overly burden the user**. Empirically, once Dice scores reach a specific level, further clicks result in only slight improvements. This indicates that **three clicks are sufficient** to cover the majority of sulcal structures effectively. This trend can already be observed in the original manuscript.

---

### Official Review · Reviewer_AYyV · 2025-10-31

**Soundness:** 2
**Presentation:** 2
**Contribution:** 2
**Rating:** 2
**Confidence:** 4

**Summary:**

The work proposes an improvement to interactive segmentation methods for cortical sulcal labeling. Given user clicks, it proposes a new prompting strategy in which click signals are propagated along surface curvature maps. This provides sulcal context to the segmentation model. The authors compare the shape-guided representation to simpler encoding methods (disk and radial distance–based) and benchmark against automatic cortical segmentation baselines. They demonstrate improved performance over simple encodings and standard baseline segmentation methods.

**Strengths:**

The paper tackles an important challenge. Registration-based cortical parcellations often lack accuracy, while newer learning-based surface segmentation methods lack substantial training data to support generalization across diverse folding patterns. This work explores an interactive improvement that reduces manual labeling effort, enabling larger and more detailed shallow sulcal parcellation maps to support better cortical analysis tools.

The motivation is clear and the proposed method is straightforward. Overall, the text is easy to follow.

**Weaknesses:**

The method is trained and evaluated on the same subset of 17 labels. Generalization to new tasks is key in interactive segmentation methods, but the paper does not evaluate performance for previously unseen regions.

For interactive tools, efficiency is also an important consideration. Yet no runtime or real-use performance stats are provided.

The baseline setup in Section 4.2 is unclear on whether these methods are retrained on the same 17 labels and data splits outlined in Section 3.3, or if they are used off the shelf. While no prior interactive sulcal-surface labeling baselines exist, numerous interactive image segmentation tools do. A comparison against a universal model such as SAM, using planar projections of sulcal and curvature maps as input, would help clarify whether a specialized approach is necessary.

The paper suggests that the chosen $\sigma$ values for the ADT and disk-based guidance signals are sufficiently small to capture fine-grained sulcal branches. However, many interactive segmentation methods perform well using single-pixel click encodings. It is not convincing that the selected $\sigma$ values are small enough, as Figure 4 shows continued improvement as ADT $\sigma$ decreases.

The introduction provides a solid overview but is overly long. The contribution could be presented far more quickly, with background details moved to a structured Related Work section.

Figures also need attention. The plots in Figure 4 are hard to interpret (some x-axis labels are missing, the scaling is not optimal, improvements are masked by stacked bars). Figure 2 introduced negative clicks, but there is no in depth methodological or experimental discussion of this.

Overall, while the problem is important, the focus on shallow sulcal regions gives the work a relatively narrow scope. Many cortical labeling tasks involve regions that span multiple sulci or gyri or rely on feature maps other than the three considered here.

**Questions:**

Line 335: "To ensure labels were constrained to sulcal regions, outputs with negative curv values were discarded" Could the authors clarify this step? It seems important but is not clearly explained

Since the same targets are used for evaluation, how does the model perform without any signal guidance at all (i.e. when trained directly for multiclass segmentation)?

In Figure 6, it seems that manual labels closely follow sulcal map borders. Instead of solving the eikonal equation with a speed function, why not simply mask the ADT or disk signal using the positive sulcal map?

---

> ### Author Response · Authors · 2025-11-27
>
> Thank you very much for the thoughtful and constructive feedback. We sincerely appreciate the time and effort you invested in evaluating our work. Some of your questions overlap with points raised by other reviewers and are addressed in our common response. We respond below to the remaining specific points of your comments.
>
> **Signal Size in Baseline Signals**
>
> Thank you for raising the concern regarding the effect of ADT and Disk signal sizes on segmentation performance. To address this, we conducted **two-sided paired t-tests comparing ADT with the smallest and largest** $\sigma$ (1/32 $\pi$ vs 1/16 $\pi$) for each of the 17 sulci across 1, 2, and 3 clicks, with **FDR correction** applied for multiple comparison correction.
>
> Our analysis shows that, while a few sulci exhibit statistically significant differences, the majority of comparisons are **not statistically significant**. For instance, with **one click**, 6 out of 17 sulci (sfs-p, imfs-h, imfs-v, pmfs-a, ds, prts) show a significant difference (p < 0.05, FDR corrected); with **two clicks**, 2 out of 17 sulci (ds, prts); and with **three clicks**, only 1 out of 17 sulci (ds). As an example, for ds with 3 clicks, the Dice scores are 87.73 $\pm$ 1.68 for $\sigma$ = 1/32 $\pi$ and 85.49 $\pm$ 2.14 for $\sigma$ = 1/16 $\pi$ (see Appendix A.3 for full results). Considering the average size of the sulcus on the cortex, the absolute difference does marginally affect the segmentation quality.
>
> These results suggest that **reducing the ADT or Disk radius does not consistently improve performance** across sulci. The t-test confirms that performance differences are negligible and not systematically biased toward either the smaller or larger signal radius. Therefore, our selected guidance signal parameters are sufficient to capture fine-grained sulcal branches while maintaining robustness.
>
> **Structure of the Introduction**
>
> We structured the introduction in a top-down manner to **clearly motivate our proposed approach.** We start from general image segmentation to orient readers who may be more familiar with standard image analysis. Next, we emphasize the limitations of fully automatic methods for identifying small and shallow sulci. Finally, we introduce our framework designed for fine-grained interactive sulcal labeling. We expect that this flow allows readers to understand **why a shape-adaptive guidance signal is needed** and how our method addresses existing limitations. While some background could alternatively be moved to a separate section, we believe that maintaining this narrative **effectively conveys the reasoning behind our method**.
>
> **Figure Layout Design & Negative Click Description**
>
> We clarify the design choices in Figure 4 in response to your concern. X-axis labels are shown only on the bottom row of each column to maintain a compact layout, and all subplots share the same y-axis range to enable **direct comparison of Dice scores** across the 17 sulci. **Stacked bars** efficiently visualize Dice score increases across click iterations and multiple guidance signal settings in **a compact layout**. To further alleviate possible interpretability issues, we have updated the Figure 4 caption to explicitly describe the axes ticks layout within the figure.
>
> Regarding negative clicks displayed in Figure 2, these are introduced to correct false positives in the predicted masks and are described in Section 2.2. In particular, we sample points from the largest mislabeled connected component in the current prediction. This component may correspond to **either a false negative or false positive region, each of which corresponds to a positive and negative click**. We note that the current description does not explicitly indicate the type of error region and its corresponding click. We have explicitly clarified these points in Section 2.2 in the revised manuscript.

---

> ### Author Response · Authors · 2025-11-27
>
> **Scope of the Study**
>
> As you have noted, some cortical labeling tasks involve regions spanning multiple sulci and gyri. Indeed, cortical parcellation methods that segment larger regions in this way are already well-supported by automatic approaches. This is because the target regions exhibit relatively consistent morphology. In contrast, **individual sulci show greater anatomical variability**. The small and shallow structures can vary in their size and location, and the number of segments for a sulcus may also differ across individuals. Such variability makes them more challenging to identify using automatic approaches.
>
> Despite their inherent variability, accurate labeling of fine-grained sulci is becoming increasingly important as recent studies are shifting toward investigation of these subtle structural differences that coarser parcellations cannot capture (Voorhies et al., 2021; Willbrand et al., 2022; Yao et al., 2023; Willbrand et al., 2023; 2024). In this context, our proposed interactive framework is specifically tailored to **support more detailed and precise sulcal labeling** and refinement. Our framework is designed to complement existing automatic approaches and facilitate fine-grained sulcal analysis.
>
> **Negative Curvature Masking**
>
> To ensure that labels remain constrained to sulcal regions, we define a sulcal region mask as the set of faces that contain **at least one vertex with mean curvature ≥ 0**. Using this mask, any output extending outside the sulcal region is thresholded. This procedure addresses retessellation artifacts and additionally ensures that **subsequent user clicks are sampled within the sulcal region**, improving the effectiveness of iterative refinement.
>
> **Model Behavior without Signal Guidance**
>
> If the interactive model receives no user guidance, it will **maintain the current prediction**, effectively producing the same output as the last iteration. When trained for multiclass segmentation instead of interactive labeling, the model performs the **same task as a non-interactive baseline**. In other words, both scenarios reflect the behavior of a standard multiclass segmentation model without additional user-provided guidance.
>
> **Alternative to Eikonal-based Propagation using Masked ADT**
>
> We experimented with training the model using the **ADT masked by the positive sulcal map** rather than solving the eikonal equation with a curvature-informed speed function. The table at the bottom shows the Dice scores for all sulci across 1-3 clicks, comparing naive ADT, masked ADT (both with $\sigma$ = 1/32 $\pi$), and WGDT (k = 10). The masked ADT **improved performance** relative to naive ADT, indicating the **importace of aligning the guidance signal** to the intended sulcal region.
>
> We note that for a single sulcus (half), the masked ADT slightly outperformed WGDT. This is likely because thresholding produces a discrete cutoff at the sulcal boundary, which incidentally favors this particular sulcal structure. Nonetheless, this **localized improvement did not generalize** to other sulci, where WGDT consistently outperformed both naive and masked ADT.
>
> To summarize, while discrete masking may occasionally benefit specific sulci, **WGDT consistently outperforms the masked equidistance-based signal** in other cases. The proposed signal captures fine-grained sulcal morphology and yields more robust, generalizable performance across diverse sulci.

---

> ### Author Response · Authors · 2025-11-27
>
> This is the table comparing Dice scores for all sulci across 1-3 clicks, comparing naive ADT, masked ADT (both with $\sigma$ = 1/32 $\pi$), and WGDT (k = 10) as described in previous comment.
>
> | Sulcus | Method | Click 1 | Click 2 | Click 3 |
> | --- | --- | --- | --- | --- |
> | cs | ADT | 97.1444 ± 0.39 | 98.4530 ± 0.27 | 98.8302 ± 0.22 |
> | cs | masked ADT | 96.5653 ± 0.37 | 98.2219 ± 0.31 | 98.5985 ± 0.27 |
> | cs | WGDT | 97.4183 ± 0.27 | 98.8403 ± 0.17 | 99.1283 ± 0.15 |
> | sprs | ADT | 85.1768 ± 0.86 | 92.7268 ± 0.60 | 94.4285 ± 0.50 |
> | sprs | masked ADT | 85.2797 ± 0.75 | 93.4862 ± 0.61 | 95.3542 ± 0.44 |
> | sprs | WGDT | 84.0327 ± 1.07 | 92.5962 ± 0.67 | 94.9712 ± 0.49 |
> | iprs | ADT | 83.5119 ± 1.30 | 90.7062 ± 1.13 | 92.8561 ± 0.99 |
> | iprs | masked ADT | 83.5126 ± 1.03 | 91.0735 ± 0.93 | 93.1073 ± 0.83 |
> | iprs | WGDT | 83.4834 ± 1.26 | 91.4273 ± 0.96 | 93.2217 ± 0.84 |
> | ifs | ADT | 78.0788 ± 1.03 | 84.9061 ± 0.94 | 88.0985 ± 0.86 |
> | ifs | masked ADT | 78.9656 ± 0.98 | 85.8582 ± 0.98 | 88.9004 ± 0.91 |
> | ifs | WGDT | 78.7493 ± 1.11 | 86.0562 ± 0.89 | 88.9338 ± 0.79 |
> | sfs-p | ADT | 75.7469 ± 0.95 | 87.5155 ± 0.90 | 90.5078 ± 0.88 |
> | sfs-p | masked ADT | 78.4933 ± 0.93 | 89.2007 ± 0.85 | 92.2189 ± 0.69 |
> | sfs-p | WGDT | 78.6678 ± 1.15 | 88.7268 ± 0.88 | 91.7244 ± 0.75 |
> | sfs-a | ADT | 81.1380 ± 0.99 | 88.9913 ± 0.92 | 91.4739 ± 0.84 |
> | sfs-a | masked ADT | 79.9322 ± 1.03 | 89.8905 ± 0.71 | 92.8096 ± 0.47 |
> | sfs-a | WGDT | 80.9677 ± 1.11 | 89.9734 ± 0.82 | 92.7759 ± 0.63 |
> | imfs-h | ADT | 78.0624 ± 1.22 | 87.8359 ± 1.04 | 91.2739 ± 0.83 |
> | imfs-h | masked ADT | 78.4661 ± 0.97 | 88.9248 ± 0.71 | 92.2204 ± 0.48 |
> | imfs-h | WGDT | 79.1803 ± 1.58 | 86.7001 ± 1.36 | 90.7158 ± 1.11 |
> | imfs-v | ADT | 70.3899 ± 1.22 | 84.0704 ± 0.91 | 88.4622 ± 0.77 |
> | imfs-v | masked ADT | 72.8597 ± 1.07 | 85.0644 ± 0.78 | 88.6736 ± 0.62 |
> | imfs-v | WGDT | 76.6719 ± 1.50 | 86.3199 ± 1.20 | 89.7889 ± 0.97 |
> | pmfs-p | ADT | 79.3308 ± 1.18 | 88.8740 ± 1.44 | 92.4081 ± 1.00 |
> | pmfs-p | masked ADT | 80.5962 ± 1.32 | 89.9501 ± 1.22 | 92.2618 ± 0.96 |
> | pmfs-p | WGDT | 88.4617 ± 1.53 | 93.2446 ± 0.99 | 94.7809 ± 0.78 |
> | pmfs-i | ADT | 72.6308 ± 1.26 | 85.8751 ± 1.32 | 88.8390 ± 1.67 |
> | pmfs-i | masked ADT | 78.3499 ± 1.66 | 87.678998 ± 1.47 | 89.3083 ± 1.40 |
> | pmfs-i | WGDT | 87.0104 ± 1.86 | 90.7017 ± 1.67 | 92.1539 ± 1.47 |
> | pmfs-a | ADT | 72.1447 ± 1.40 | 85.4625 ± 1.31 | 89.2603 ± 1.34 |
> | pmfs-a | masked ADT | 75.3482 ± 1.54 | 86.7888 ± 1.32 | 90.5261 ± 1.17 |
> | pmfs-a | WGDT | 81.5941 ± 1.98 | 88.1575 ± 1.58 | 90.7940 ± 1.34 |
> | ds | ADT | 68.2744 ± 1.89 | 82.4209 ± 1.90 | 85.4899 ± 2.14 |
> | ds | masked ADT | 74.7038 ± 1.96 | 85.9303 ± 1.64 | 88.8126 ± 1.31 |
> | ds | WGDT | 82.0982 ± 2.11 | 90.1754 ± 1.52 | 92.1105 ± 1.19 |
> | ts | ADT | 68.7187 ± 1.58 | 81.0904 ± 1.96 | 86.2846 ± 1.75 |
> | ts | masked ADT | 72.6365 ± 1.78 | 84.3816 ± 1.80 | 87.3019 ± 1.60 |
> | ts | WGDT | 81.2000 ± 1.71 | 90.06997 ± 1.12 | 92.3218 ± 0.90 |
> | aalf | ADT | 66.6842 ± 1.30 | 82.0596 ± 1.43 | 86.8091 ± 1.41 |
> | aalf | masked ADT | 76.2131 ± 1.49 | 85.8818 ± 1.60 | 88.1607 ± 1.44 |
> | aalf | WGDT | 82.8784 ± 1.61 | 89.3874 ± 1.57 | 90.0531 ± 1.54 |
> | half | ADT | 79.5412 ± 1.12 | 87.0194 ± 0.99 | 89.6780 ± 0.75 |
> | half | masked ADT | 85.6054 ± 1.05 | 90.9517 ± 0.92 | 92.4014 ± 0.78 |
> | half | WGDT | 83.3178 ± 1.35 | 89.4717 ± 1.10 | 90.4334 ± 1.00 |
> | prts | ADT | 74.1229 ± 1.34 | 87.4529 ± 0.97 | 91.1665 ± 0.82 |
> | prts | masked ADT | 77.6051 ± 1.19 | 88.3401 ± 0.99 | 90.6484 ± 0.77 |
> | prts | WGDT | 82.4369 ± 1.73 | 90.4779 ± 1.37 | 92.9589 ± 1.09 |
> | lfms | ADT | 74.8064 ± 1.10 | 87.9253 ± 1.28 | 91.6779 ± 1.35 |
> | lfms | masked ADT | 79.4440 ± 1.55 | 89.1694 ± 1.35 | 92.3417 ± 1.08 |
> | lfms | WGDT | 84.2006 ± 2.17 | 91.0810 ± 1.67 | 93.6942 ± 1.25 |

---

### Author Response · Authors · 2025-11-27

We thank all the reviewers for the detailed and constructive feedback and for the time and effort invested in evaluating our work. We address the common concerns and questions raised across the reviewers below.

---

> ### Author Response · Authors · 2025-11-27
>
> **Training Protocol Clarification & Generalization beyond the LPFC Sulci**
>
> We thank the reviewers for raising this important point. The LPFC is relatively enlarged and slowly maturing part of the human brain, with complex cortical folds. As illustrated in Figure 1, it contains **sulci with distinct morphological characteristics**, including size, branching pattern, and overall shape. While addressing generalizability to other cortical regions is also important, we focused on the LPFC because it encompasses a wide variety of sulcal morphologies. This makes it a rigorous testbed for assessing our guidance signal’s robustness across different types of sulci.
>
> These sulcal characteristics pose challenges for integrating all sulci into a single model without losing such region-specific details. Consistent with common practices in medical image interactive segmentation (Wang et al., 2018; Luo et al., 2021; Diaz-Pinto et al., 2022), where models are often trained separately for specific organs or regions, **we trained a separate supervised model for each sulcus**. This approach allows the network to specialize in the characteristics of each individual sulcus. For every experimental configuration, this produces one learned model per sulcus and ensures the evaluation covers a wide spectrum of anatomical patterns within the LPFC.
>
> While a few recent works have explored multi-object interactive segmentation (Rana et al., 2023; Yue et al., 2024), extending single-instance interactive segmentation to multiple objects is **non-trivial**. In a multi-object scenario, **the number of possible user click sequences increases**, making click simulation more complex. Users can focus on different sulci **in an arbitrary order**, which makes predicting the next interaction difficult. Also, training a model to achieve **balanced performance** across multiple sulci is challenging. Given these complexities, we focus on single-sulcus binary segmentation, which allows us to clearly demonstrate the effectiveness of the proposed shape-adaptive guidance signal.
>
> In summary, we recognize the importance of ensuring that our method generalizes across diverse cortical folds. Building on the motivations described above, our **per-sulcus training** and **evaluation on the LPFC** provides a structured way to examine how well the guidance signal handles **distinct sulcal patterns**. We have revised the corresponding sections in the manuscript (Section 2.1 and 5) to clarify the training protocol and avoid any ambiguity in how the per-sulcus models are constructed and assessed.
>
> **References**
>
> Amit Rana, Sabarinath Mahadevan, Alexander Hermans, and Bastian Leibe. Dynamite: Dynamic query bootstrapping for multi-object interactive segmentation transformer. In ICCV, 2023.
>
> Yuanwen Yue, Sabarinath Mahadevan, Jonas Schult, Francis Engelmann, Bastian Leibe, Konrad Schindler, and Theodora Kontogianni. AGILE3D: Attention Guided Interactive Multi-object 3D Segmentation. In International Conference on Learning Representations (ICLR), 2024.

---

> ### Author Response · Authors · 2025-11-27
>
> **Runtime and Real-use Performance**
>
> We appreciate the reviewers’ comment regarding efficiency and real-world performance, which is indeed important for interactive tools. We measured the runtime of our proposed method under the standard experimental setup. Specifically, our measurement was performed on **intel Xeon 6526Y** and **NVIDIA RTX 6000 Ada Generation**, which is the same environment for model training and evaluation. We report the elapsed time of WGDT signal encoding, the retessellation between the sphere-mapped cortical surface and the 40962-vertex sphere of icosahedral subdivision for both input features and output segmentation, and the model forward pass:
>
> - **Signal encoding:** approximately 200 ms per click
> - **Retessellation:** approximately 200 ms in total
> - **Model inference:** approximately 27 ms per forward pass
>
> Therefore,  a full interaction cycle (signal encoding + retessellation + model inference) is completed **well under one second**, enabling real-time feedback during annotation.
>
> We also agree that real-world user studies are valuable for assessing practical annotation speed and usability. Nevertheless, conducting user-based studies for every method with different sulci in our dataset would **be highly time-consuming and difficult to scale**. Each annotation must be performed by trained neuroanatomy experts rather than general users. This requirement substantially increases the methodological burden.
>
> To provide a systematic and reproducible evaluation within our relatively small cohort size (**N=72**), we first aimed to mitigate potential bias that may arise from specific initial click locations. As stated in section 4.2, we generated diverse initial click locations for each sulcus to reduce sensitivity to any single initial click location. Specifically, 10 initial click points were sampled within each manual sulcal label, selected to maximize both their distance from the label boundary and mutual separation. This strategy enabled us to approximate a wider range of interaction patterns within a limited dataset.
>
> Regarding the user interface, we are developing a **GUI prototype** to support click-based interactions on cortical surface using our model, aiming for further refinement and potential release in the future. We have added additional section in the results (Section 4.3) to report these runtime measurements and revised the corresponding section (Section 3.3) to clarify our simulation-based evaluation in the manuscript.
>
> **Automatic Sulcal Labeling Baseline Configuration**
>
> To clarify the setup of our **automatic (non-interactive) baselines**, we note that all of the baselines were originally developed for fine-grained sulcal labeling (Lyu et al., 2021; Lee et al., 2025a; Lee et al., 2025b). These methods directly target the automatic labeling of large and small cortical sulci. Each model was retrained on our dataset using the same geometry features (curv, sulc, inflated.H) and hyperparameter settings reported in their respective publications.
>
> By retraining the baselines on the same dataset, we ensured a **fair comparison** between interactive and non-interactive approaches. This guarantees that the observed differences in performance reflect the **effectiveness of our guidance encoding and interactive framework**, rather than differences in training data or label distribution.

---

> ### Author Response · Authors · 2025-11-27
>
> **SAM-based Baseline Consideration**
>
> Regarding a comparison with a universal segmentation model such as SAM, we note several **fundamental challenges** in constructing a fair baseline for sulcal labeling using planar projections:
>
> - **Planar Projection Issues:** Projecting a sphere onto a plane requires at least one cut on the surface, as a planar domain cannot represent the sphere’s genus-0 topology without discontinuities. As a consequence, mapping the cortical surface from the sphere to a plane inevitably introduces **distortions and discontinuities**. Mean curvature, average convexity, and sulcal branching patterns are distorted especially near poles or boundaries. These distortions compromise local feature fidelity and continuity, making a single projection or even multiple patch-based projections an imperfect approximation of the true 3D structure.
> - **Modality Mismatch:** SAM was trained on natural RGB images with photometric and texture priors. Geometry features such as mean curvature or convexity have very **different statistical distributions** from photometric signals. Simply feeding these features into SAM, even with a front-end adapter, may not produce meaningful embeddings, and retraining or fine-tuning parts of SAM would require substantial architectural changes and introduce additional training instability.
> - **Unestablished Methodology:** To our knowledge, no prior work has proposed a principled SAM-based approach for cortical sulcal labeling. Creating an artificial baseline would require novel methodological development beyond the scope of this work. Attempting this could risk **unfair or misleading comparisons**, as such a baseline may underperform for reasons unrelated to the effectiveness of our interactive guidance strategy.
>
> In summary, due to **topological distortions, modality mismatch, and the lack of an established procedure**, a SAM-based baseline is currently not feasible in a way that would fairly evaluate the contributions of our shape-adaptive guidance signal.
>
> **Sensitivity to Curvature Noise & Atypical Brains**
>
> Our experiments are based on **well-reconstructed cortical surfaces**, and the sulcal labeling scenario assumes data obtained from healthy participants. For pathological brains, sulcal labeling may be inherently undefined. In **excessively noisy subjects**, which are outside the intended scenario, the curvature-based speed function of the WGDT signal may not operate reliably. It should be noted that manual labeling would also be challenging under these conditions. Although such extreme cases may pose challenges, our proposed guidance signal can handle **a wide variety of small and variable sulcal morphologies**, enabling accurate labeling with only a few clicks per sulcus.

---

### Author Response · Authors · 2025-12-03
**Rebuttal Summary**

Dear AC,

We appreciate the effort you are dedicating to the evaluation, especially given the recent changes in the reviewing process. In response to the reviewers’ comments, we have carefully addressed their concerns for our work through additional experiments, clarifications, and revisions in the corresponding sections of the manuscript.

To support your decision-making process, we summarize the main reviewer concerns and our corresponding responses, and we believe our revisions have signficantly improved the manuscript. We mark all changes in blue in the revised manuscript.

---

> ### Author Response · Authors · 2025-12-03
> **Strengths**
>
> The reviewers consistently **highlighted several key strengths of the work**.
>
> - All three reviewers agreed that the work is **well-motivated by the challenges** of labeling shallow sulci and the limitations of automatic methods [AYyV, XRiF, njFB].
> - Also, they acknowledged that the proposed approach **reduces manual labeling effort** while enabling more accurate and detailed sulcal labeling, even **for small and challenging structures**.
> - The reviewers found that the proposed guidance signal design is **straightforward** [AYyV] and **attractive for sulcal analysis** [njFB] by adapting to anatomical structure of cortical folds with **thorough formulation** [XRiF].
>
> While these strengths were appreciated, the reviewers also raised several concerns, all of which we believe have been fully addressed in the discussions and revisions.

---

> ### Author Response · Authors · 2025-12-03
> **Common Concerns**
>
> - One common concern raised by all three reviewers [AYyV, XRiF, njFB] was the **generalization of the method** beyond LPFC sulci as its applicability to other cortical regions. LPFC was chosen because it is one of the most variable cortical regions (Voorhies et al., 2021; Lyu et al., 2021). Hence, its **complex and highly diverse sulcal morphology** provides an ideal context for evaluating our method. As already discussed in the introduction, **obtaining manually annotated sulcal labels itself is resource-intensive** and requires the expertise of neuroanatomists. The scarcity of such annotations constrains the set of cortical regions for which comprehensive evaluation is currently feasible. **Per-sulcus modeling** allows each model to **specialize for the distinct characteristics of individual sulci**. We have updated this rationale in **Sections 2.1** and **5**.
> - Another shared concern involved the **rationale for per-sulcus modeling** [AYyV, XRiF, njFB]. Although technically feasible, we emphasize that multi-sulcus interactive segmentation could introduce challenges such as numerous possible click sequences, shifts in user focus, and difficulties in balanced training as also discussed in several multi-object interactive segmentation approaches (Rana et al., 2023; Yue et al., 2024). In **Section 2.1**, we have clarified that training a separate supervised model for each sulcus allows specialization for each structure which aligns with common practices in interactive medical image segmentation. .
> - **Runtime and real-use performance** was another common concern [AYyV, XRiF]. We have reported the **measured time per initial click**, including **signal encoding**, **re-tessellation**, and **model forward pass**, showing that a single click requires less than **0.5 seconds**. These results support real-time interactive use and were incorporated into **Section 4.3** for additional runtime analysis.
> - Another common concern was the potential **comparison with a universal segmentation model** such as SAM [AyYV, XRiF]. While the suggestions are valuable, constructing a fair SAM-based baseline is currently not feasible for serveral reasons. Planar projections of the spherical data inevitably require cutting of the data, which further introduces **large polar distortions** depending on the chosen location of the cut. Also, SAM was trained on RGB images and **cannot directly interpret geometric features (curvature or convexity)**. Finally, no established SAM-based methodology exists for sulcal labeling. Attempting such an inadequately established baseline could produce misleading comparisons.
> - Finally, the reviewers raised concerns regarding **the noise in curvature and atypical brains** [XRiF, njFB]. Like many cortical surface labeling studies, we assume the normal brain surface, which is already highly variable. In atypical brains, curvature-based propagation may be affected, but accurate surfaces must be reconstructed first as a prerequisite for accurate sulcal labeling. While smoothing may help mitigate the noise effect, we emphasize that this issue is not unique to our method; many other cortical surface processing tasks (e.g., parcellation) are similarly affected. These have been clarified in **Section 5**.

---

> ### Author Response · Authors · 2025-12-03
> **Reviewer-Specific Concerns & Summary**
>
> - **Reviewer AYyV’s** main concern was the **narrow scope of our study**. While the reviewer noted that many cortical labeling tasks involve regions spanning multiple sulci or gyri (i.e., parcellation), this perspective completely **overlooks the complexity of sulcal labeling**. Parcellation tasks are relativley simple as they assign labels to a fixed number of regions, whereas sulcal labeling is **inherently more challenging** due to the varying number of segments in sulcal anatomy as also shown in Fig. 6. In fact, automatic parcellation methods already achieve over 86% accuracy without user interaction (Zhao et al., 2019; Parvathaneni et al., 2019; Ha & Lyu, 2022). This suggests that our problem is far more difficult and **cannot be easily reduced to the parcellation problem**. Our interactive framework is designed to complement these complex tasks using the proposed shape-adaptive guidance.
> - **Reviewer AYyV** also raised issues regarding the **size of equidistance-based signals** (ADT/Disk) and the **clarity in the manuscript description**. We have performed additional statistical tests showing that reducing the radius **does not consistently improve performance** in the ADT/Disk approaches. **Section 2.2** is now revised to clarify how positive and negative clicks refine the current prediction. **Figure 4** captions have been updated for better clarity of its axes tick layout, and baseline setups are now clarified in **Section 4.2**.
> - **Reviewer XRiF** raised concerns about **WGDT parameter selection**, **justification of the curvature-based speed function**, and **saturation of accuracy** with additional clicks. We had already clarified that WGDT parameters are empirically chosen in Appendix A.1 of the original manuscript, and noted that learning-based approaches could further reduce the hyperparameter search range in **Section 5**. The curvature-based speed function is justified by surface geometry characteristics; mean curvature was selected to account for the morphology of small sulci. Additional interactions are unlikely to provide substantial improvement after a certain number of clicks, based on the trends observed in **Section 4.1**. This reasoning justifies our choice of simulated clicks.
> - **Reviewer njFB** noted the **small cohort size** in the evaluation. As discussed, the acquistion of manual sulcal labels is a labor-intensive task that requires trained neuroanatomists. Due to the infeasibility of large-scale data acquisition, the small dataset size may introduce a bias in evaluation as some clicks could favor certain methods. To mitigate this potential bias, we simulated **diverse initial click locations per sulcus** as already described in **Section 3.3** to account for variable user interaction in the evaluation given the limited dataset size. As the reviewers have highlighted as one of the key strengths, we have shown our proposed WGDT signal consistently achieved **higher accuracies regardless of the initial user click pattern** compared to other baseline signals.
>
> In summary, the reviewers provided constructive feedback on generalization, parameter choices, and methodological details. They consistently recognized the **novelty**, **motivation**, and **practical utility of our approach**. By providingclarifications, additional analyses, and detailed rationale, we believe that we have fully resolved all of the raised issues.

---

### Meta-Review · Area_Chair_8iAJ · 2026-01-06

**Summary:**

While the reviewers acknowledge the importance of the problem tackled and the effectiveness of the proposed solution in reducing the manual effort, they expressed concerns related to the scope (generalization) of this work, the baselines used for comparison, the runtime of the method, the sensitivity to hyper-parameters, and the presentation of some parts of the work.

**Reviewer Concerns:**

The concerns related to the baseline comparisons, runtime, and presentation issues are reasonably convincingly addressed by the rebuttal. The sensitivity to hyper-parameters, however, is essentially left as future work. Furthermore, the question of scope remains debatable; considering the nature of this work, a biomedical-focused venue would seem like a better choice in terms of target audience.

**Reviewer Scores:**

Although some of the reviewers' concerns have been convincingly addressed by the authors' rebuttal, important ones remain; in particular the scope of the study, which was mentioned by all reviewers. As such, the AC expects that the majority of the reviewers would have maintained a score below the acceptance threshold. The AC recommends the authors to resubmit their work to a venue that is more focused on the biomedical nature of their work, where they are likely to reach a larger audience.

---

### Decision · Program_Chairs · 2026-01-26

Reject